# A wheat CC-NBS-LRR protein Ym1 confers WYMV resistance by recognizing viral coat protein

Yiming Chen [1,7], Dehui Kong[1,7], Zongkuan Wang [1,7], Jiaqian Liu[2], Linghan Wang[1], Keli Dai[1], Jialun Ji[1], Wei Chen [3], Xiong Tang[1], Mingxing Wen[1], Xu Zhang[1], Huajian Zhang[1], Chengzhi Jiao[1], Li Sun[1], Haiyan Wang[1], Xingru Fei[4], Hong Guo[4], Bingjian Sun[5], Xiaorong Tao [6], Wei Wang [1], Jian Yang [2], Xiue Wang [1] ✉ & Jin Xiao [1] ✉

*Ym1* is the most widely utilized gene for wheat yellow mosaic virus (WYMV) disease control in worldwide wheat breeding. Here, we successfully isolated the responsible gene for *Ym1*. It encodes a typical CC-NBS-LRR type R protein, which is specifically expressed in root and induced upon WYMV infection. *Ym1*-mediated WYMV resistance is likely achieved by blocking viral transmission from the root cortex into steles, thereby preventing systemic movement to aerial tissues. Ym1 CC domain is essential for triggering cell death. Ym1 specifically interacts with WYMV coat protein, and this interaction leads to nucleocytoplasmic redistribution, a process for transitioning Ym1 from an auto-inhibited to an activated state. The activation subsequently elicits hypersensitive responses and establishes WYMV resistance. *Ym1* is likely introgressed from the sub-genome $X^n$ or $X^c$ of polyploid *Aegilops* species. The findings highlight an exogenous-introgressed and root-specifically expressed *R* gene that confers WYMV resistance by recognizing the viral component.

Viruses are the acellular and obligate pathogens that cause a wide range of plant diseases, posing a great threat to crop production. The mechanisms of plant-virus interactions are complex and sophisticated, mainly due to the presence of virus-transmission vectors[1,2]. Viruses recruit a series of host proteins to accomplish the complex infection cycle[1,2], including viral particle disassembly[3,4], viral translation[5], formation of viral replication complex[6], virion assembly, cell-to-cell virus movement[7,8], and long-distance virus transporting[8]. Plants have developed two-tier defense mechanisms to prevent or limit the damage caused by viral infection. The first defense layer involves DNA methylation[9], RNA silencing (also known as RNA interference or RNAi)[10,11], m⁶A RNA modification[12], autophagy[13], post-translational modification[14,15], phytohormones[16], and production of reactive oxygen species[17] or other viral defense-related secondary metabolites[18]. To counter plant defenses, viruses deploy multifunctional viral effectors to effectively hijack key component hubs, such as the geminivirus AC5 protein for reducing DNA methylation[19]; viral suppressors of RNA silencing (VSRs) for suppressing RNAi[20]; BSMV γb protein for compromising autophagy and salicylic acid (SA)-mediated defense responses[21,22]. To cope with these pathogenic effectors, plants activate the second defense layer by recognizing viral effectors through nucleotide binding-leucine rich repeat (NBS-LRR, or NLR) resistance (R) proteins to initiate robust hypersensitive responses (HR).

[1]National Key Laboratory of Crop Genetics & Germplasm Enhancement and Utilization, Nanjing Agricultural University/Zhongshan Biological Breeding Laboratory/CIC-MCP, Nanjing, Jiangsu, China. [2]Institute of Plant Virology, Ningbo University, Ningbo, Zhejiang, China. [3]State Key Laboratory of Pharmaceutical Biotechnology, School of Life Sciences, Nanjing University, Nanjing, China. [4]Yandu District Agricultural Science Research Institute, Yancheng, Jiangsu, China. [5]College of Plant Protection, Henan Agricultural University, Zhengzhou, Henan, China. [6]Department of Plant Pathology, Nanjing Agricultural University, Nanjing, Jiangsu, China. [7]These authors contributed equally: Yiming Chen, Dehui Kong, Zongkuan Wang. ✉e-mail: xiuew@njau.edu.cn; xiaojin@njau.edu.cn

Many NBS-LRR-type plant *R* genes have been identified and proved to confer dominant resistance against virus pathogens[23]. Such plant R proteins can recognize/interact with a virus component, directly or indirectly to initiate defense responses. The *Potato virus X* (PVX) coat protein (CP) acts as an elicitor, which can be recognized by potato NBS-LRR type R protein StRx1. This recognition disrupts the intramolecular interaction between the LRR and CC-NB-ARC domains of StRx1, leading to conformational change for resetting a nucleotide-bound state of the NB-ARC domain[24]. In addition to CP, the viral movement protein, replicase protein, and RNAi suppressor protein all can act as potential elicitors[25]. The plausible arm race between plants and virus pathogens promotes rapid genome evolution in both sides, adding complexity to understand the plant defense mechanisms against viral pathogens. In recent years, significant progresses have been achieved toward the understanding of host defense strategies against several insect-vectored viral pathogens[26]. In contrast, the molecular mechanisms of plant defense against soil-borne viruses are still very poor, which, at least partially due to the difficulty in establishing an efficient plant-virus-soil-inhabiting vector artificial inoculation system under laboratory conditions.

Global wheat production is severely threatened by viral pathogens including the soil-borne bymovirus wheat yellow mosaic virus (WYMV). The obligate root-habitat plasmodiophorid *P. graminis* (*Pg* hereinafter) is the vector for WYMV infection[27]. Under suitable environment conditions, *Pg* fungus produces primary and secondary mobile zoospores to colonize cereal roots where the WYMV particles are injected[27]. In susceptible plants, the viral particles are transported from roots to stems and to the newly emerging young leaves for multiplication. Eventually, the infected wheat plants show symptoms such as yellow-striped mosaic leaves and stunted growth. WYMV threats more than 2.2 million square hectometers of Chinese wheat grown areas annually, causing 30% up to 70% yield reduction[28]. The soil-borne *Pg*-vectored WYMV is hardly to control by conventional integrated pest management strategies, leaving deployment of host resistance the only practical option for efficient disease control.

The WYMV genome consists of bipartite positive-sense RNAs, RNA1 and RNA2. RNA1 (~7.6 Kb) encodes a large polyprotein (~270 kDa) that will be processed into eight mature proteins including P3, 7 K, CI, 14 K, VPg, NIa-Pro, NIb, and CP[29]. In addition, a small open reading frame (ORF) termed PIPO overlaps with the P3-coding region[30]. RNA2 (~3.5 Kb) encodes a polyprotein (~100 kDa) that will be cleaved into two mature proteins: P1 and P2[29]. The viral proteins play various roles during viral infection including virus replication (7 K, CI, 14 K, VPg, CP, NIb, and P2), intercellular movement (P3, CI, 14 K, and PIPO), polyprotein translation, cleaving and processing (CP, NIa-Pro and VPg), endoplasmic reticulum export (14 K), viral particle assembly (CP) and RNA-silencing suppression (P1)[31]. The viral RNA sequences have been used for RNAi in host to block viral infection. Transgenic plants expressing the antisense coding region of viral *NIb* gene endue durable WYMV resistance in wheat[32]. Disruption of intracellular host factors, which are essential for virus replication, transcription, or movement, also results in recessive resistance against virus infection in plant. The knocking out of recessive resistance genes, i.e., *TaLIP*[33], *TaPDIL5-1*[34], *TaeIF4e*[35], *TaMTB*[12], and *TaVTC2*[36] enhanced WYMV resistance in wheat. In another study, a wheat papain-like cysteine protease *TaRD21A* confers WYMV resistance by releasing a small signal peptide[37]. Nevertheless, natural host resistance provides a more robust defense against WYMV infection and can be used directly in crop breeding. To date, 14 WYMV resistance (*R*) genes or quantitative trait loci (QTL) have been identified in wheat, which are located on eight chromosomes (wheat chromosomes 2A, 2DL, 3BS, 5AL, 6DS, 7A, 7BS, and *Dasypyrum villosa* chromosome 4VS)[31]. So far, only *Ym2* on 3BS has been cloned. *Ym2*, originated from *Aegilops sharonensis*, encodes a typical CC-NBS-LRR R protein. *Ym2* is specifically expressed in roots

and confers WYMV resistance possibly by preventing WYMV movement from fungal vector into plant roots[38].

The major-effect QTL for WYMV resistance located on 2DL is the most widely utilized resistance source in wheat breeding of China. The resistance allele is present in different varieties, and thus is assigned different names such as *QYm.nau-2D*[39], *YmYF*[40], *YmIb*[41], *Ym1*[42], *YmMD*[43], and *Q.Ymym*[29,44]. In previous study, we propose the above named genes have a common origin: being introgressed from a wild relative species into common wheat[39]. The recombination suppression between alien introgression fragment and its counterparts of wheat makes very difficult for map-based cloning of the locus. Here, we report the map-based cloning of *QYm.nau-2D* (*Ym1* hereinafter) by promoting the homoeologous recombination. We predict a CC-NBS-LRR type R protein encoding gene is the candidate gene for *Ym1*. *Ym1* transcription is specifically induced in roots. The knocking down or knocking-out of *Ym1* compromises WYMV resistance, while overexpression of *Ym1* enhances WYMV resistance in wheat. Additionally, we found Ym1 recognizes and interacts with the WYMV CP. Ym1-CP interaction alters the nucleus-cytoplasmic localization of Ym1, and ultimately triggers HR. The work provides insights into the molecular nature of *Ym1*-mediated host resistance against a soil-borne fungus vectored virus, and also provides a tool for developing durable resistance against WYMV in wheat breeding.

## Results
### Fine mapping of the WYMV resistance locus *Ym1*
In our previous study, we predicted the resistance allele of *Ym1* is an alien introgression from the genome N of *Aegilops uniaristata*. *Ym1* region corresponds to ~11.2 Mb physical region of chromosome 2DL of wheat Chinese Spring (CS in brief hereinafter)[39]. To overcome homoeologous recombination between the alien fragment and its wheat homeologous 2D counterpart, a double hybrid $F_1$ cross scheme was designed using Yining Xiaomai (YNXM in brief hereinafter, the donor of *Ym1*), 2011I-78 (WYMV susceptible), and CS *ph1b* (*pairing homologous gene 1b*) mutant (WYMV susceptible). Plants that are heterozygous for *Ym1* and homozygous for the mutant gene *ph1b* were identified (Supplementary Fig. 1) using two *ph1b* diagnostic markers *Xwgc2111* and *Xwgc2049* and *Ym1* diagnostic marker *SSR_X3* (Supplementary Data 1). The derived 326 $BC_1F_2$ individuals heterozygous for *Ym1* locus was genotyped using two flanking markers *InDel_M41* and *InDel_M412* for recombinant screening. Three homozygous recombinant haplotypes (N, O, and P) were identified. WYMV resistance evaluation in field nursery and by WYMV titer detection using Reverse Transcription-Polymerase Chain Reaction (RT-PCR) indicated haplotype O was susceptible, and haplotypes N and P were resistant (Fig. 1A, B and Supplementary Fig. 2). Thus, *Ym1* was narrowed down into a physical interval flanked by two markers, *2ESTK2* and *InDel_FA192*. Our previous study showed WYMV susceptible variety CS doesn't carry *Ym1*, and WYMV resistant wheat variety 'Fielder' (the sequenced variety) carries *Ym1*[39]. The mapped *Ym1* region corresponds to 5.6 Mbp (600.1 to 605.7 Mbp) in CS 2DL, in which 73 genes are annotated; while corresponds to 610.7 ~ 616.5 Mb in Fielder 2DL, in which 65 genes are annotated (Supplementary Data 2). The above annotated genes in the two genomes were used for reciprocal BLAST sequence alignment. Sixteen annotated genes were uniquely present CS, in which five are NLR type genes, *TraesCS2D01G510000*, *TraesCS2D01G510100*, *TraesCS2D01G510300*, *TraesCS2D01G511600*, and *TraesCS2D01G511700*. Eleven annotated genes were uniquely present in Fielder, in which only *TraesFLD2D01G565300* is a NLR type gene.

### *Ym1* candidate gene encodes a typical CC-NBS-LRR R protein
RNA-seq was used for *Ym1* candidate gene prediction, using samples of YNXM leaf and root tissues collected from WYMV-infected and

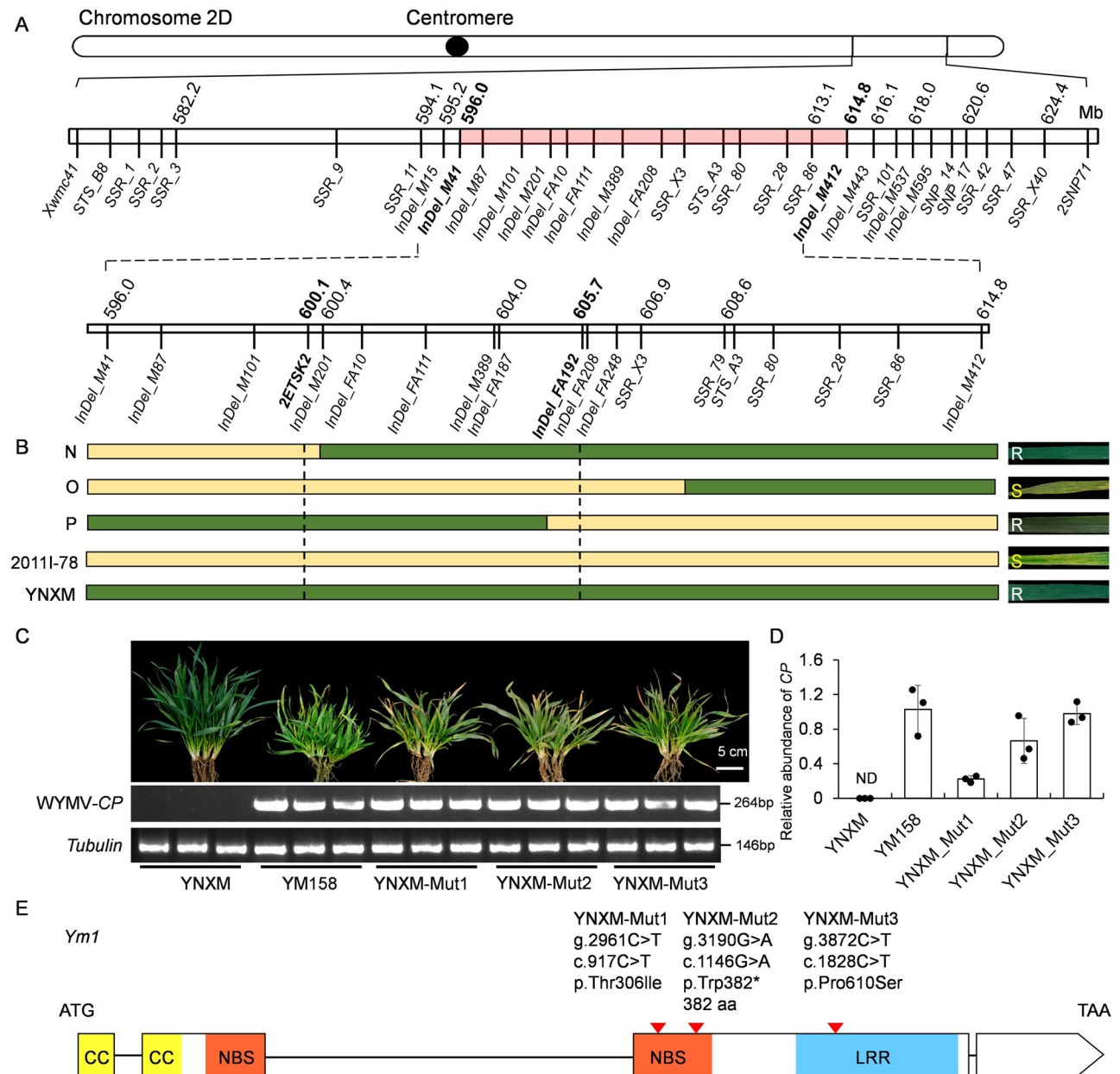

**Fig. 1 | Map-based cloning of WYMV resistance locus *Ym1*. A** Genetic and physical maps of *Ym1* region on chromosome 2DL. The region in pink represents the putative alien introgression fragment. Markers in bold are used to screen recombinants (upper) and define the finely mapped region, respectively. **B** Recombinant types (designated as N, O, and P) and their WYMV responses. S/WYMV+: WYMV susceptible with abundant coat protein (CP) transcripts in leaves; R/WYMV-: WYMV resistant with no-detectable CP transcript in leaves. **C** WYMV symptoms (upper) and CP transcripts revealed by RT-PCR (lower) of Yining Xiaomai (YNXM), Yangmai 158 (YM158), and three susceptible YNXM mutants (YNXM-Mut1, YNXM-Mut2 and YNXM-Mut3). The wheat *Tubulin* gene was used as control. **D** CP transcripts quantification in the leaves of YNXM, YM158, and three YNXM susceptible mutants by qRT-PCR. ND not detectable. The values of qRT-PCR are the mean ± SD (*n* = 3 biologically independent experiments). **E** Diagram showing the nucleotide and amino acid *Ym1* mutations in the three WYMV susceptible mutants YNXM-Mut1, YNXM-Mut2 and YNXM-Mut3.

WYMV-free field plots. Thirty-six out of the 65 Fielder annotated genes within the *Ym1* 5.8 Mb region were expressed. Among the differentially expressed genes (DEGs) in response to WYMV infection, seven were downregulated and no gene was upregulated in leaf tissue; eight were upregulated and eight were downregulated in root tissue (Supplementary Fig. 3). WYMV is a soil-borne virus, and this makes us to consider eight root-specifically upregulated DEGs as candidate genes for *Ym1*. Among the six upregulated DEGs, *TraesFLD2D01G565300* is uniquely present in Fielder and the only one be annotated as an NLR type R protein.

We generated WYMV susceptible mutants by **Ethyl methane sulfonate (EMS)** mutagenesis of YNXM. Three independent WYMV susceptible mutants were identified and verified by the WYMV *CP* RNA abundance analysis using RT-PCR and quantitative RT-PCR (qPCR) (Fig. 1C, D). The three mutant lines were re-sequenced and the sequences were aligned to the reference genome sequence of Fielder. Ten annotated genes located in the *Ym1* region were found to have nucleotide mutations in at least one mutant. Among them, six genes had mutations in their introns causing no splice-site changes, and the remaining four genes (*TraesFLD2D01G564000, Traes FLD2D01G564500, TraesFLD2D01G565300,* and *TraesFLD2D01G5*

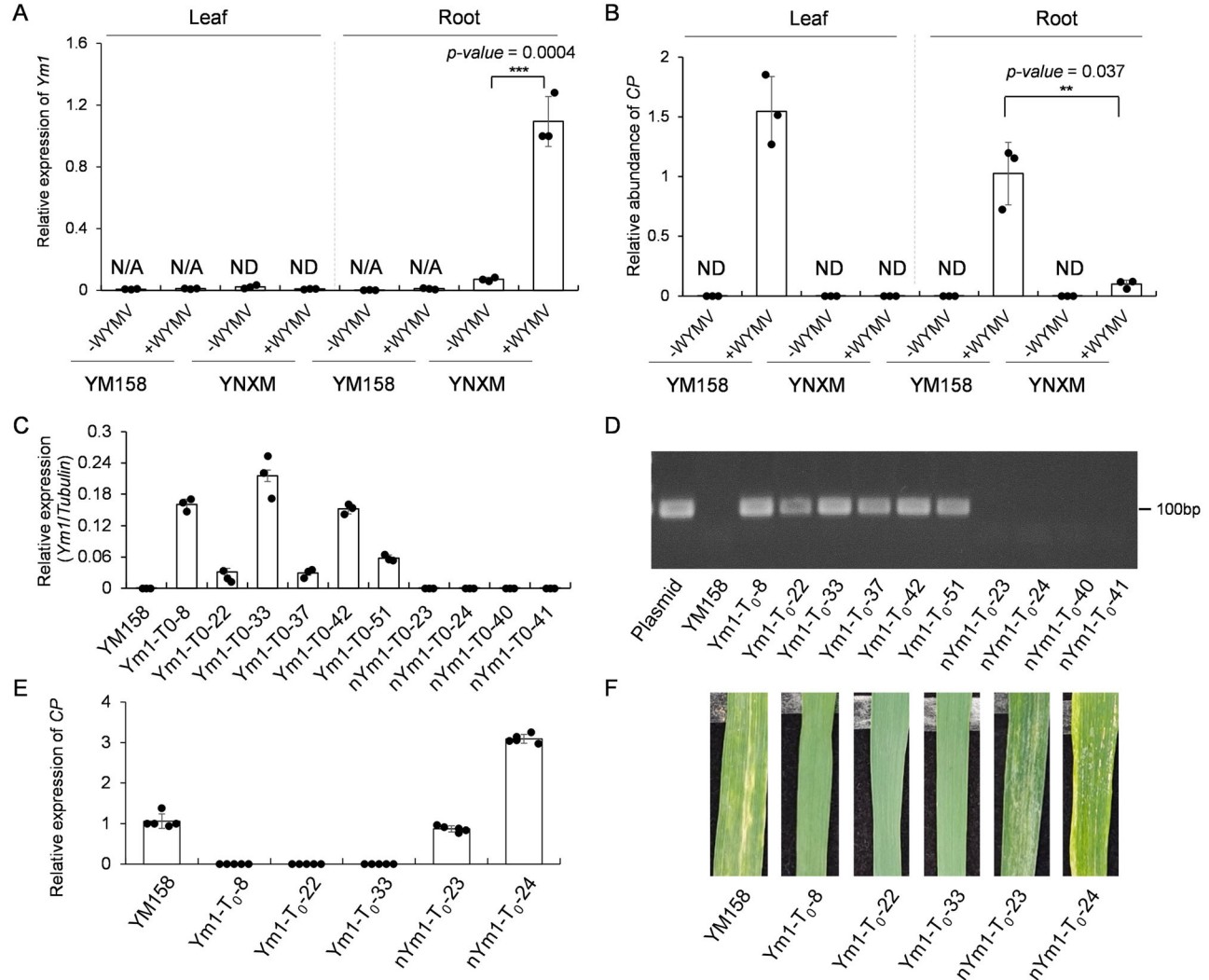

**Fig. 2 | Over-expression of *Ym1* in WYMV susceptible variety Yangmai 158 enhanced its resistance. A, B** Wheat *Ym1* gene expression (**A**) and WYMV *CP* gene transcript detection (**B**) in roots and leaves of Yining Xiaomai (YNXM, resistant) and Yangmai 158 (YM158, susceptible) grown in WYMV-infected (+WYMV) nursery and WYMV-free field (-WYMV). Each data is presented as means ± SD (*n* = 3 biologically independent plants). ****P* < 0.001, ***P* < 0.01, **P* < 0.05 based on two-tailed *t*-tests. N/A not applicable; ND not detectable. **C, D** Identification of *Ym1* $T_0$ transgenic plants

in YM158 by qRT-PCR (**C**) and PCR (**D**). Results represent the means ± SD (*n* = 3 technical replicates). ND: not detectable. **E, F** Quantification of *CP* transcript abundance by qRT-PCR (**E**) and disease symptoms observation after artificial inoculation of WYMV clone (**F**) for three Ym1 positive transgenic plants and two negative transgenic plants in YM158. YM158 was used as the control. Results represent the means ± SD (*n* = 5 technical replicates).

*65500*) had nonsense or missense mutations in their exons (Supplementary Data 3).

Notably, all the three susceptible mutants had exon mutations in *TraesFLD2D01G565300*, which was uniquely expressed and upregulated in the root of YNXM upon WYMV infection. YNXM-Mut1 and YNXM-Mut3 are missense mutants (both with C to T mutation at 2961 bp and 3872 bp of genomic DNA that led to amino acid changes of threonine (Thr) to isoleucine (Ile) and proline (Pro) to serine (Ser), respectively). YNXM-Mut2 is a nonsense mutant (with G to A mutation at 3190 bp of genomic DNA that leads to a premature stop codon and 382 aa truncated protein) (Fig. 1E and Supplementary Fig. 4). Thus, we consider *TraesFLD2D01G565300* as the putative candidate for *Ym1*. *TraesFLD2D01G565300* encodes an incomplete CDS (1426 bp), starting with ATG but not ending up with a stop codon (TGA, TAG or TAA). We predicted the full-length CDS of *TraesFLD2D01G565300* based on YNXM's RNA-seq data, and cloned its full-length cDNA (3201 bp) and genomic DNA (gDNA, 5281 bp) from YNXM and Fielder. The full-length gDNA has four exons and three introns, encoding a typical CC-NBS-LRR R protein of 1066 amino acid (aa) residues (Fig. 1E).

## Functional validation of *Ym1* in WYMV resistance

*Ym1* transcript was specifically expressed and upregulated in roots of YNXM after WYMV infection, but was undetectable in roots of WYMV susceptible variety Yangmai 158 (YM158 for short hereafter) regardless of the infection (Fig. 2A). YM158 grown in natural WYMV nursery had no difference in virus accumulation in both leaves and roots (Fig. 2B). Six independent positive transgenic plants were identified by over-expressing *Ym1* in YM158 (Fig. 2C, D). Three positive plants (Ym1-$T_0$-8, Ym1-$T_0$-22, and Ym1-$T_0$-33) and two negative plants (nYm1-$T_0$-23 and nYm1-$T_0$-24) were inoculated with WYMV infectious clone. None of the positive plants had obvious WYMV symptom and detectable viral *CP* transcript in their leaves. In contrast, the two negative plants were not significantly different from susceptible receptor YM158 for their WYMV symptom and *CP* transcript abundances (Fig. 2E, F). These data validated *Ym1* candidate gene plays a positive role in WYMV resistance.

We next validated the function of *Ym1* by knocking down *Ym1* in YNXM by virus-induced gene silencing (VIGS). BSMV:*Ym1*-*CC* and BSMV:*Ym1*-*LRR* were used, respectively targeting the CC and LRR domain of *Ym1*, BSMV-EV was used as the negative control. In the roots

of BSMV:Ym1-*CC* or BSMV:Ym1-*LRR* inoculated plants, *Ym1* transcripts were significantly decreased at 21 days post inoculation (dpi) (Supplementary Fig. 5A). The increased WYMV susceptibility and accumulated virus *CP* transcripts in *Ym1* knocked-down plants suggested that knockdown of *Ym1* compromised WYMV resistance in YNXM (Supplementary Fig. 5B, C).

We further knocked out *Ym1* in Fielder by CRIPSR-Cas9-based editing. Two *Ym1* targeting sites were designed, with target 1 at the CC domain and target 2 at the LRR domain (Fig. 3A). Eleven independent $T_0$ plants were obtained and four of them, namely CRYm1-4-2, CRYm1-6, CRYm1-8, and CRYm1-1-1 carrying 2 bp up to 3.2 Kbp nucleotide deletions resulted in premature stop codons and truncated Ym1 protein (Fig. 3A). $T_2$ plants carrying homozygous mutation alleles (based on Sanger sequencing) were derived from each four lines were WYMV susceptible under natural infection in field nursery (Fig. 3B), and WYMV *CP* transcript accumulation was detected in their leaves (Fig. 3C). The spatial expression of *Ym1* was further examined by RNA in situ hybridization (using *Ym1* 3' terminal fragment as an antisense probe (Supplementary Data 1). *Ym1* transcript could not be detected in roots of CRYm1-4-2, but could be detected mainly distributed in root endodermis tissues of wild Fielder (Fig. 3D). This demonstrates *Ym1*-conferred WYMV resistance depends on its specific expression in the roots.

## *Ym1* confers WYMV resistance by blocking virus invasion other than fungal colonization in the root

When grown in WYMV-infected nursery, *CP* accumulation in the leaves could be detected in WYMV susceptible YM158 but not in WYMV resistant YNXM. Whereas, *CP* accumulation in roots could be detected in both YM158 and YNXM, with the *CP* abundance in YNXM was significantly lower than that in YM158 (Fig. 2B). We assume that *Ym1* may function in preventing viral invasion in root. To validate this, we investigated whether lower viral load in YNXM root tissues is due to the blocking of *Pg* colonization or virus invasion. *Pg* was artificially inoculated and fungal abundances were compared at 14 days after *Pg* inoculation. The four genotypes, despite of the *Ym1* presence (Fielder, *Ym1* OE transgenic plants) or absence (*Ym1*-edited mutant lines, YM158), had no difference in *Pg* abundance, either in root or rhizosphere soil (Supplementary Fig. 6A, B). This suggest that *Ym1*-conferred WYMV resistance is not associated with the prevention of *Pg* colonization.

The effect of *Ym1* on virus invasion was investigated by RNA in situ hybridization using WYMV *CP* as an antisense probe (Supplementary Data 1). The viral distribution in root and basal stem tissues of Fielder and *Ym1*-edited mutant line CRYm1-4-2 grown in the WYMV nursery were compared. In WYMV-resistant Fielder, *CP* was restricted only to the outer layer of root, including epidermis, cortex and endodermis. The virus failed to invade into the stele and further move up to the basal stem. While in WYMV susceptible CRYm1-4-2, *CP* could be detected both in root cortex layer and stele, as well as in vascular tissues of the basal stem (Fig. 3E). This suggest that the knock-out of *Ym1* in Fielder compromised its WYMV resistance by promoting virus infection in the roots. We performed viral transmission assay by infiltration into *Nicotiana benthamiana* leaves. When infiltrating WYMV infectious clone alone, CP could be detected surrounding the infiltration site; while when co-infiltrating WYMV infectious clone together with Ym1-GFP construct, the detected CP abundance was obviously reduced and CP distribution was restricted at the infiltration site (Supplementary Fig. 7). These suggest root-specific expressed *Ym1* confers WYMV resistance by inhibiting viral movement in the roots and subsequently transporting to the stems.

## Ym1 interacts with WYMV CP to trigger hypersensitive response (HR)

We investigate whether *Ym1* confers WYMV resistance by direct recognition of viral proteins. We screened potential elicitor in *Ym1*-triggered immune responses using yeast two-hybrid (Y2H) assay to test interactions between WYMV-encoding proteins and full-length Ym1 or its truncated sequences, including CC (residues 1 aa to 157 aa with the CC domain), NBS (residues 137 aa to 503 aa, containing the NBS domain), LRR (residues 471 aa to 1066 aa containing the LRR domain), CC-NBS and NBS-LRR (Fig. 4A). The CP was the only WYMV-encoding protein which had weak interaction with full-length Ym1 and strong interaction with Ym1's LRR domain (Fig. 4B and Supplementary Fig. 8A). Co-immunoprecipitation (Co-IP) and split luciferase complementation imaging (LCI) assays validated their interactions (Fig. 4C, D). Y2H assay indicated Ym1 CC domain had self-interaction (Supplementary Fig. 8B). Generally, the NLR type R proteins function in disease resistance by triggering cell death, possibly via their CC domains. Cell death assay in *N. benthamiana* leaves indicated, at 48 h after infiltration, only transiently expressed CC domain triggered cell death. Infiltration of the full-length *Ym1* alone could not trigger cell death (Fig. 4E), while co-infiltrating full-length *Ym1* and *CP* could trigger cell death (Fig. 4F). Since the cell death symptom induced by Ym1-CP interaction was weaker compared to CC and Bax, trypan blue staining was conducted to validate the result (Fig. 4F). This suggested that CP may act as the elicitor of Ym1 for triggering cell death, which was further validated in wheat roots. Our results showed that co-infiltration of full-length *Ym1* and *CP* triggered weaker cell death than that when infiltrating CC domain, hinting a possible inhibition of CC triggered cell death in full-length Ym1. We artificially inoculated WYMV infectious clones into roots of different wheat genotypes. The cell death was detectable only in roots of *Ym1*-carrying genotypes Fielder and *Ym1* transgenic plants, while was undetectable in roots of *Ym1* knockout line CRYm1-4-2 and YM158 (Fig. 4G), demonstrating the elicitation of cell death is dependent on the recognition of CP by Ym1.

We investigated whether the WYMV susceptibility of the mutants were due to the loss of cell death. The mutated *Ym1* genes in YNXM-Mut1 and YNXM-Mut3 did not affect their interaction with CP, however, co-infiltrating mutated *Ym1s* with *CP* could not trigger cell death (Supplementary Fig. 9A, B). The mutated *Ym1* in YNXM-Mut2 encodes a truncated protein and failed to interact with CP, as a result, co-infiltration of this mutated *Ym1* and *CP* could not elicit cell death. These results further demonstrated Ym1 recognizes WYMV CP, elicits cell death and hinders viral invasion.

Nucleocytoplasmic distribution is reported to be required for NLR activation, for example in the case of StRx1[45], MLA10[46], and SNC1[47]. In our study, we transiently expressed fusion protein Ym1-GFP which was subcellularly localized in cytoplasm and nucleus (Fig. 5A). However, when simultaneously delivering Ym1-GFP and HA-tagged CP constructs into *N. benthamiana* leaves, GFP signals were only detected in the cytoplasm not in the nucleus (Fig. 5A). Such a localization change of Ym1-GFP by co-expressing CP was validated by anti-GFP immunoblotting (Fig. 5B). We propose that Ym1 interacted with the invaded CP, and their interaction led to Ym1's nuclear exporting into the cytoplasm, which may be pivotal for the activation of Ym1.

To test the above speculation, we examined the role of nuclear exporting of Ym1 in cell death triggering. As expected, the Ym1-GFP-NES and CC-GFP-NES (adding a nuclear export signal to C terminus) was uniquely localized in the cytoplasm while the Ym1-GFP-NLS and CC-GFP-NLS signals (adding a nuclear export signal to C terminus) was uniquely localized in the nucleus (Fig. 5C and Supplementary Fig. 10). Cell death was triggered only when co-infiltrating Ym1-GFP-NES and CP or infiltrating CC-GFP-NES; no cell death was observed when co-infiltrating Ym1-GFP-NLS and CP or infiltrating CC-GFP-NLS (Fig. 5D and Supplementary Fig. 10). These data revealed that cytoplasmic-localization of Ym1 is required for cell death triggering, which may be essential for its function in WYMV resistance.

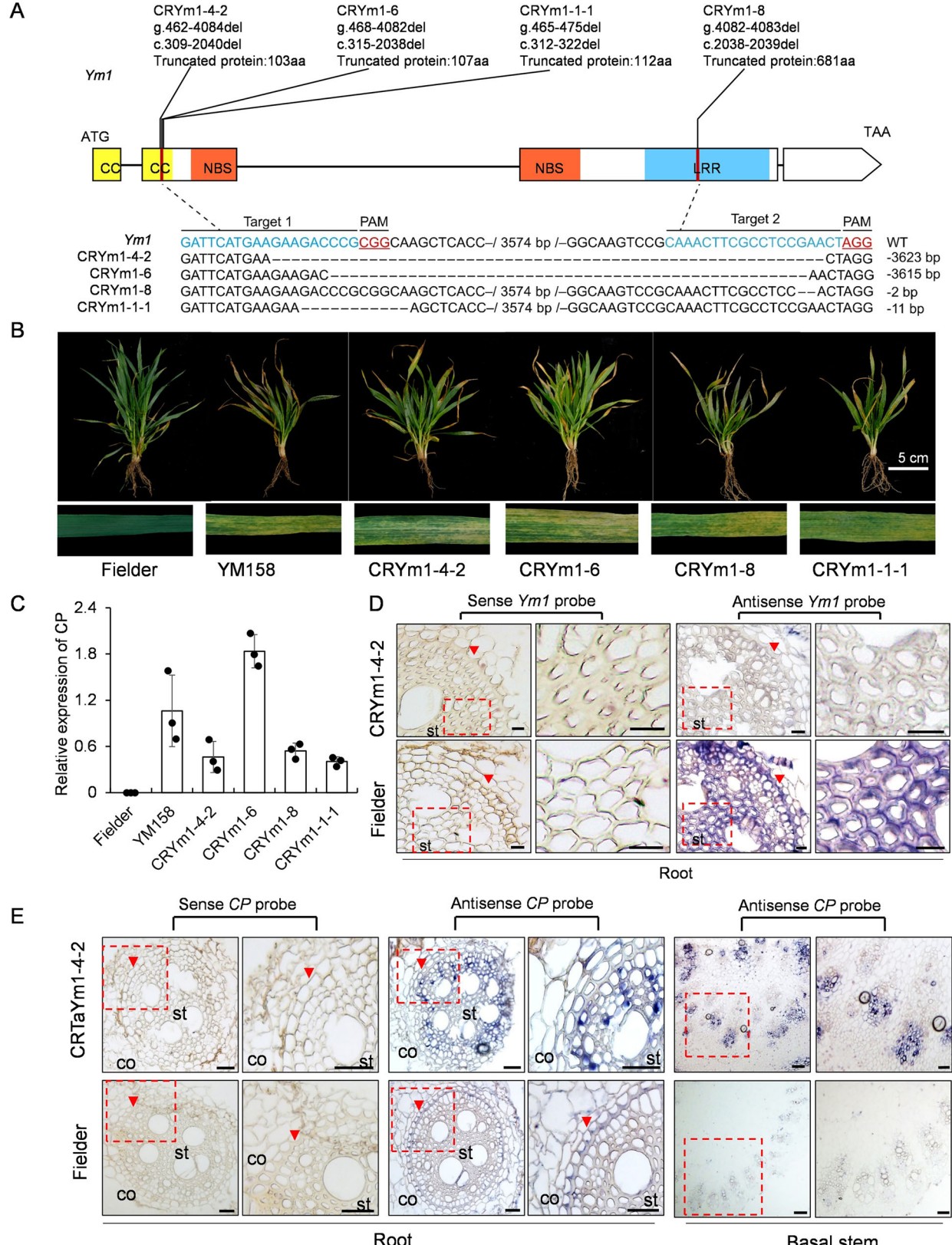

## The sub-genome X of polyploid *Aegilops* species is predicted to be the direct donor of *Ym1*

The *Ym1* in WYMV resistant YNXM and Fielder are identical, while *Ym1* is absent in WYMV susceptible CS. Based on the nucleotide sequence variation (SV) of WYMV resistant and susceptible genotypes, we developed a *Ym1*-specific marker *Ym1-DM* to validate its association with WYMV resistance (Supplementary Data 1). A diversity panel of 372 wheat varieties[39] were genotyped using *Ym1-DM*, and the specific amplicons were uniquely detected in 106 WYMV resistant varieties, which are proposed to carry *Ym1*.

To deduce the origin of *Ym1*, we constructed a phylogenetic tree based on the sequences of Ym1 and 76 CNL-class NLR proteins from

**Fig. 3 | Targeted gene editing of *Ym1* in Fielder compromised WYMV resistance. A** Diagrams showing names, positions of deletions in gDNA and cDNA, and corresponding amino acid residues of truncated proteins of the four *Ym1* CRISPR-Cas9-edited $T_1$ mutants in Fielder background. Targets for *Ym1* gene editing are shown by vertical red lines. The sgRNA and the protospacer-adjacent motif (PAM) are highlighted in blue and red, respectively, in the wild type (WT) sequences, the deleted nucleotides are shown as "-", and the numbers of deleted base pairs of each line are shown on the right. **B** WYMV disease symptoms of seedlings (upper) and leaves (lower) of *Ym1* gene-edited mutant plants. **C** Abundance of WYMV CP transcripts in the leaves of *Ym1*-edited plants grown in the WYMV nursery. The wheat *Tubulin* gene is used as the internal control. Values are means ± SD (*n* = 3 biologically independent plants). ND not detectable. **D** Visualization of *Ym1* in transverse root by RNA in situ hybridization using *Ym1* fragment at 3′terminal as a probe. The arrowheads point to the endodermis. The images enclosed in the dashed frames (left) are shown in an enlarged view on the right. st stele, co cortex. Scale bars = 20 μm. **E** Visualization of WYMV *CP* RNA in transverse root and basal stem sections from RNA in situ hybridization using WYMV *CP* as the probe. The arrowheads indicate the endodermis. The images enclosed in the dashed frames (left) are shown in an enlarged view on the right. st stele. co cortex. Scale bars = 50 μm.

Triticeae and related genera (Supplementary Data 4). Ym1 is most closely related to Yr10 (wheat stripe rust resistance protein) and Pi36 (rice blast resistance protein), but their amino acid sequence identities were low (with only 46% and 40% bootstrapping support, respectively) (Supplementary Fig. 11), revealing the distinctness of Ym1 from previously characterized plant NLR proteins. Ym1 is far away from the recently cloned WYMV resistance protein Ym2[38], confirming their different origins. The microsynteny of the *Ym1* region (~5.6 Mbp in the Fielder genome) in different diploid, tetraploid, and hexaploid Triticeae species were analyzed. We found *Ym1* orthologs were only present in *Secale cereal* (two accessions) and *D. villosum* (one accession) (Supplementary Fig. 12), with 72.2%, 72.2%, and 73.3% sequence identities, respectively.

In an early study, we proposed *Ym1* in YNXM was situated in an alien introgression from a genus *Aegilops* species[39]. We re-sequenced two most possible donor species for *Ym1*, i.e., diploid *Ae. umbellulata* (UU)[39] and tetraploid *Ae. columaris* (U$^c$U$^c$X$^c$X$^c$). The generated sequence reads were aligned to sequence of Fielder chromosome 2D. In the mapped *Ym1* region, total reads hit in tetraploid *Ae. columaris* was higher than that in diploid *Ae. umbellulata* (Fig. 6A). We propose such difference is due to the presence of X$^c$ in *Ae. columaris*, which is proposed to be the donor of introgressed *Ym1* region.

We attempted to clone the *Ym1* homoeologous alleles from *Aegilops* diploid species carrying genomes U, M, N, and C, but all failed. We cloned or searched *Ym1* orthologs from 18 *Aegilops* species and performed phylogenetic analysis (Supplementary Data 5). The *Ym1* had 89.25% ~ 99.68% sequence identities with its orthologs from the analyzed *Aegilops* species (Fig. 6B). The two *Ym1* orthologs in some polyploidy species were designated as hap-1 and hap-2. Notably, *Ym1* had >99% sequence identities with orthologs from *Ae. columaris* (U$^c$U$^c$X$^c$X$^c$), *Ae. neglecta* (U$^n$U$^n$X$^n$X$^n$) and *Ae. neglenta* (U$^n$U$^n$X$^n$X$^n$N$^n$N$^n$, hap-1), which contain X$^c$ or X$^n$ subgenomes. *Ym1* had ~92.3% sequence identities with orthologs from *Ae. crassa*, *Ae. vavilovii* and *Ae. juvenalis*, which carry the X$^{cr}$ genome. This indicated genomes X$^c$ and X$^n$ are more closely related, while X$^{cr}$ is more distinct from X$^c$ or X$^n$[48]. We obtained *Ym1* orthologs in seven diploid, one tetraploid and three hexaploid *Aegilops* species containing S genome and hexaploid *Aegilops* species containing N$^n$ genome, but they all had relatively lower similarities with *Ym1*, being 89.25% ~ 93.15%. The function of *Ym1* orthologs in *Ae. columnaris*, *Ae. neglecta* and *Ae. vavilovii* was analyzed by artificial inoculation of WYMV infectious clones into the roots of the three species. The expression of the three ortholog genes was induced in roots after WYMV inoculation, while *CP* transcript was undetectable in their leaves. This indicated the *Ym1* ortholog genes from the three species have similar expression pattern as that of *Ym1*, which may be associated with WYMV resistance (Supplementary Fig. 13). To conclude the above results, we propose X$^n$ or X$^c$ in tetraploid/hexaploid *Ae. neglenta* and tetraploid *Ae. columaris* is most likely the direct donors of *Ym1*.

## Discussion

Plant viruses pose major constraint to optimum crop yield. WYMV infection may cause up to 70% yield loss of wheat. To counteract viral infection, plants exploit their genetic sources of reliable resistance, such as *R* genes mediated resistance. The resistance gene/QTL on chromosome 2DL have been repeatedly identified in different wheat genotypes. Our previous study revealed *Ym1* in YNXM, introgressed from alien species, confers high WYMV resistance. However, due to recombinant suppression of alien introgression with its wheat counterpart, our efforts in cloning *Ym1* has been unsuccessful. In this study, we promoted recombination and narrowed down the genetic region by introducing wheat pair homologous gene mutant gene *ph1b*. Taking the advantage of released genome sequence of wheat variety Fielder (carrying *Ym1*), EMS-based mutagenesis and RNA-seq, we isolated *Ym1* candidate gene. *Ym1* encodes a nucleotide binding-leucine rich repeat (NBS-LRR) R protein. The function of Ym1 in WYMV resistance is validated by the generation of transgenic plants overexpressing *Ym1* and knocking-down or knocking-out plants by VIGS or genome editing. *Ym1* is root-specific expressed and confers resistance by direct recognition of viral CP, triggering cell death and blocking the viral invasion in roots. For soil-borne viral diseases, the root cells are conceivably a battlefield for the plant resistance genes to prevent virus invasion. The entry of WYMV into wheat is initiated by the attachment of *Pg* zoospores to root cells, followed by the penetration with a developed tubular structure for the injection of zoospore contents into the cytoplasm of the root cell. The viruliferous zoospores penetrate the root epidermal cell, develop into a multinucleate sporangial plasmodium in root outer cells (such as epidermal, cortical and endodermal cells)[3,49]. Once the virus enters the cells, it multiplies and moves cell-to-cell by employing the host cellular systems such as intercellular plasmodesmata connections and subsequently loads into plant vasculature for systemic infection[50]. *Ym1* is root specifically expressed, transcriptionally induced by WYMV infection, while its gene expression was hardly detected in the above ground tissues (stem and leaf). In another study, when the leaves of resistant variety Madsen harboring the same resistance locus *Ym1* is mechanically inoculated, WYMV could be detected in the leaf and stem tissues but not in the root tissues[51], highlighting the root-specific expression of *Ym1* for its function in disease resistance. The fungal vector *Pg* colonizes the roots irrespective of host WYMV resistance status[51]. The virus replicates inside the *Pg* vector cells[52]. We found *Ym1* functions in effectively blocking WYMV invading rather than preventing the *Pg* fungal colonization, and this explained why we detected a relatively lower viral load in roots of YNXM. The recently cloned *Ym2*[38] also encodes a CC-NBS-LRR (CNL), showing similar spatial expression pattern in roots. *Ym2* is proposed to confer WYMV resistance by inhibiting the ingress of virus from *Pg* into host root cell, indicating Ym2 and Ym1 have common resistance mechanism. In our study, *Ym1* functions by recognizing virus CP which requires the access of virus into host cells. However, we did not observe Ym2-CP direct interaction by Y2H. We propose the two *R* gene may employ different resistance pathways, which makes gene pyramiding for durable resistance possible in breeding program for WYMV resistance.

To date, most of the identified *R* genes (*StRx1*, *Sw-5*, *Tm-2*, etc.) show dominant resistance by triggering antiviral responses via direct or indirect recognition of viral effector molecules (which may differ for

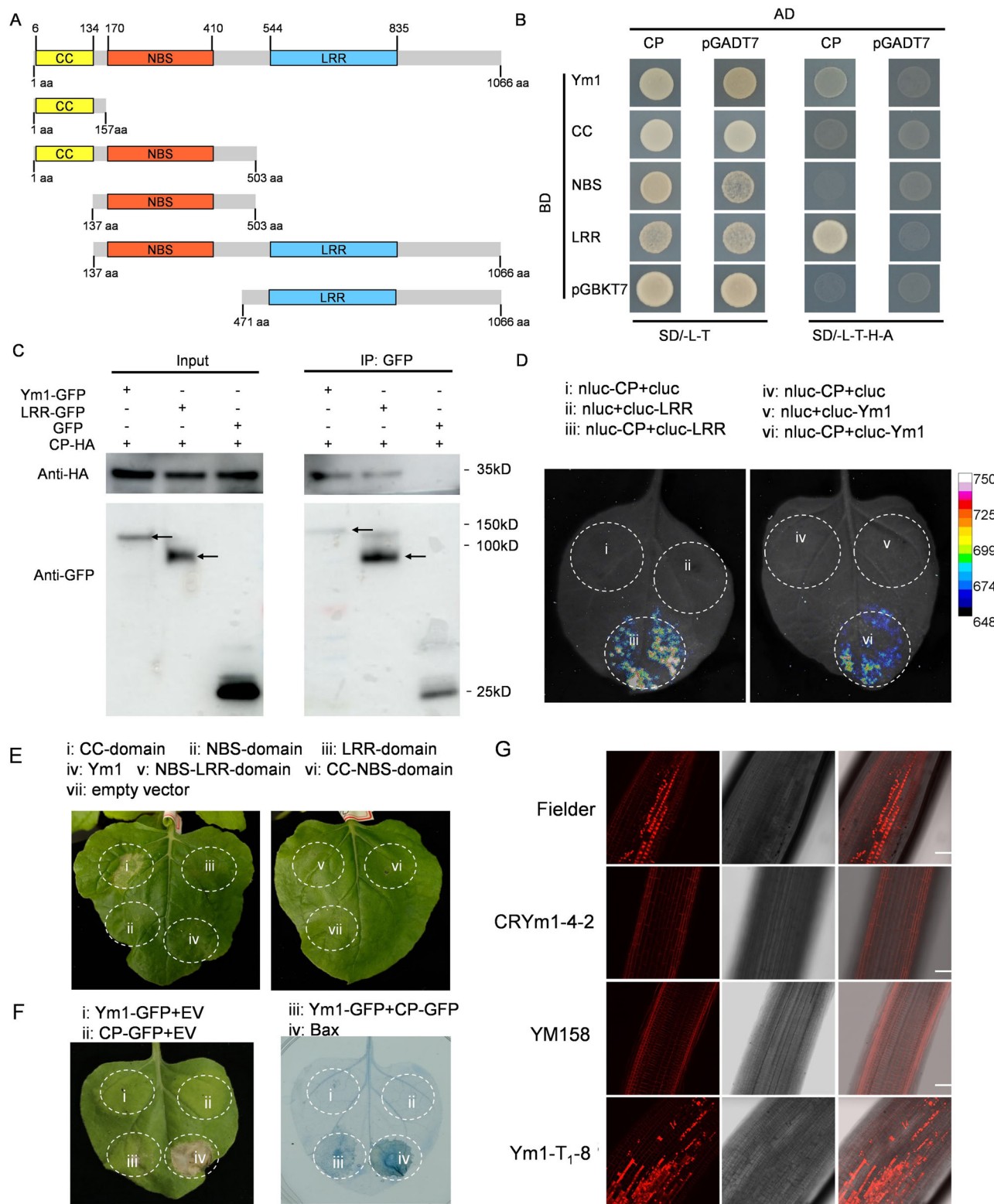

different virus-host systems)[23]. The viral CPs, which are involved in almost every stage of the viral infection cycle[53], represent pathogen molecules in many cases and are specifically recognized by the R proteins for triggering immune responses[23]. In the most extensively characterized *StRx1*- PVX interaction system, potato StRx1 recognizes PVX CP to trigger downstream defense responses[54]. Multiple domains of StRx1 coordinated together through the intramolecular and intermolecular interactions to convert the recognition of virus elicitor CP

into signal initiation event. In the absence of CP, StRx1 LRR domain interacted with the CC-NBS, and CC domain interacted with the NBS-LRR[55]. The intramolecular interaction retains the StRx1 molecule in the auto-inhibited state, which is disrupted in the presence of CP[55]. The recognition executed by LRR domain results in a perturbed binding between the LRR and ARC, a process required for signal initiation[56]. Ym1 is a typical CNL R protein and also functions by recognizing WYMV CP. We postulate Ym1 employs a similar resistance mechanism as

**Fig. 4 | Ym1 directly interacted with WYMV CP to trigger cell death. A** Diagrams showing the constructs of full-length Ym1 and its truncated versions. The truncated Ym1 domains were fused to the C-terminal of the Green fluorescent protein (GFP) tag and expressed in N. benthamiana leaves by *Agrobacterium*-mediated transient transformation. Grey: complete Ym1 protein; Yellow: CC domain; Red: NBS domain; Blue: LRR domain. The numbers indicate the starting or stopping amino acid (aa) sites of the tested domains. **B** Y2H assays revealing the critical domains for WYMV CP-Ym1 interaction. The corresponding sequences were fused to the Gal4 DNA-binding domain (BD); the WYMV CP was fused to the Gal4 activation domain (AD). The yeast co-transformed with BD and AD constructs were plated onto synthetic dropout (SD) media lacking Leu and Trp (left) and SD media lacking Ade, His, Leu, and Trp (right). Images were taken at 3 days after incubation. **C** Co-immunoprecipitation (Co-IP) analysis showing the interactions between Ym1-GFP fusion protein or its truncated domain LRR-GFP fusion protein with WYMV CP in N. benthamiana leaves. Total proteins and IP products were detected by Western blotting using anti-HA or anti-GFP antibodies at 48 hpi. The arrows point the bands for Ym1-GFP (upper) and LRR-GFP (lower). The experiment was repeated three times independently with similar results. **D** LUC interaction of Ym1 or its truncated LRR domain with WYMV CP detected by split luciferase complementation imaging (LCI) assay on N. benthamiana leaves. The leaf areas infiltrated with *Agrobacterium* cultures expressing nluc-CP+cluc, nluc+cluc-Ym1, and nluc+cluc-LRR were used as negative controls, respectively. Luciferase activity was captured using an imaging apparatus at 2 days post inoculation. **E** Cell death triggering activity assay of Ym1 and its truncated domains or domain combinations. Each image was taken at 96 h post-inoculation (hpi). **F** Phenotypic observations of Ym1-mediated WYMV resistance elicited by WYMV CP. Triggered hypersensitive response (HR) cell death was observed in N. benthamiana leaves when co-expressing Ym1-GFP and CP-HA combinations, but not observed when co-expressing the remaining tested combinations. Fusion proteins were expressed in *N. benthamiana* leaves by Agro-infiltration. The trypan blue staining was conducted to validate the cell death result. **G** Cell death assay of different genotypes in wheat roots after artificial inoculation of WYMV infectious clones. Bars = 50 μm. The experiment was repeated three times independently with similar results.

potato StRx1, since Ym1 also uses LRR to interact with CP for signaling transduction. However, unlike StRx1 which induces cell-death signaling through its NB domain[57], Ym1 functions through its CC domain.

StRx1 confers extreme resistance (ER) in vivo by preventing viral replication without triggering necrotic cell death[58,59]. When the PVX CP was expressed from a transgene, StRx1 also initiate HR-type cell death[59]. In our results, Ym1 elicited cell death in recognition of CP in *N. benthamiana* leaves. Because of a lack of artificial inoculation system using virus contaminated-*Pg* vector under laboratory condition, we cannot determine whether *Ym1*-mediated WYMV resistance relied on HR or ER.

The function of R proteins in plant defense is closely associated with their proper subcellular localization[60]. The barley MLA[58], tobacco N[61], *Arabidopsis* RPS4[62], and potato StRx1[45] all needs to be distributed in both cytoplasm and nucleus compartments for full functionality. The cytoplasmic localization required for cell-death induction was also demonstrated for MLA10, Sr33, and Sr50 in fungal disease resistance[58]. Ym1 is localized to cytoplasm and nucleus, while Ym1 and CP interaction leads to a change in nucleocytoplasmic distribution. The nucleocytoplasmic distribution of several NB-LRR proteins depends on their binding to chaperone complex which is both cytoplasm and nucleus localized[58,62–65]. The chaperone complex, consisting of SGT1, RAR1, and HSP90, is known to function in proper NB-LRR protein folding and stabilizing as a potentially signaling-competent state. Silencing of *SGT1* impaired the accumulation of StRx1 protein in the nucleus[45]. We will next investigate whether Ym1 also uses this piggyback mechanism that endues its capacity in both cytoplasmic and nuclear localization.

Ran GTPase-activating protein 2 (RanGAP2) has been described as a co-factor and is essential for activation of the immune response against PVX. RanGAP2 interacts with the StRx1 CC domain through its N-terminal WPP domain that determines its localization outside of nuclear envelope[65–67]. As a cytoplasmic retention factor, RanGAP2 facilitates the nucleocytoplasmic partitioning of StRx1 by sequestrating it into cytoplasm[67]. The cytoplasm accumulated NES fused Ym1 CC domain triggered HR, while nucleus accumulated NLS fused Ym1 CC failed to do it, implying a proper nucleocytoplasmic distribution of Ym1 is pivotal for immune response activation and defense signaling. We propose, instead of binding to the chaperone complex for the autoinhibitory state, Ym1 may interact with other cofactors for its activation after the recognition of the elicitor CP.

Naturally occurring *R* gene-mediated disease response against viruses is a reliable and durable source of resistance. Gene losses are pervasive during the genome evolution of a given species. The genes that are not essential or dispensable are to some degree readily to be lost because their losses have no or slightly impact on fitness or benefit to organisms, at least under certain circumstances[68]. This is the so-called "less is more" hypothesis[69]. In our research, *Ym1* and its orthologs are only present in very few proportions of *Aegilops* species containing $X^c$, $X^n$, $X^{cr}$, $S^*$, $N^n$ genomes but absent in most of the investigated species including C, D, U, M, N lineages. *Ym2* was also derived from the wild species *Ae. sharonensis*[38]. The root specifically expressed and WYMV induced *Ym1* is likely to be "dispensable" and lost in the areas where are not suitable for the WYMV epidemic. Gene loss was also found for another WYMV resistance *Ym2*, the susceptible allele (*ym2*) of which has a deleterious mutation (1 bp deletion) at gene coding region that cause loss-of-function mutation[38]. Compared to the domesticated wheat, tremendous genetic diversity is preserved in wild relatives[70]. The loss of gene in cultivated species might be occurred randomly but can be regained from its relative species. The clone of *Ym1* provides an example of the introgression of genomic diversity from *Aegilops* species into cultivated common wheat for WYMV resistance.

In summary, we map-based cloned *Ym1* by overcoming the recombination suppression. The candidate gene, *Ym1*, encodes a typical CNL R protein. We present a working model to illustrate the role of *Ym1* in WYMV resistance (Fig. 7). *Ym1* transcription is activated specifically in roots upon WYMV infection. Ym1 triggered plant defenses to block viral invasion without affecting root colonization of *Pg*. Ym1 CC domain is essential for triggering cell death. Without viral infection, Ym1 is present as auto-inhibited status possibly via intra-molecular interaction. When sensing the viral invasion, Ym1 recognizes and interacts with WYMV CP, leading to cytoplasmic localization and changing Ym1 into an activated status. The activated Ym1 triggers hypersensitive response (HR), and prevents the viral invasion into root stele (Fig. 7). Our findings unveil a defense model of R gene elicited by CP in wheat-WYMV interaction system.

## Methods
### Plant materials
For fine mapping of *Ym1*, a BC$_1$F$_2$ segregating population consisting of 326 individuals was developed employing a double hybrid F$_1$ scheme (Supplementary Fig. 1) using three wheat varieties: WYMV resistance donor variety YNXM, and two susceptible lines 2011I-78, and the Chinese Spring *Ph1b* mutant (CS*ph1b*). 2011I-78 were kindly provided by Dr. J. Dvorak of the University of California, Davis, USA. Plants that are homozygous *ph1bph1b* (5B$^{ph1b}$/5B$^{ph1b}$) and heterozygous *Ym1* (2D$^{YNXM}$/2D$^{Aet}$) were selected using two *ph1b* diagnostic markers *Xwgc2111* and *Xwgc2049*, and *Ym1* linkage markers. Plant materials used in this study also includes 119 accessions of 30 *Aegilops* species, in which 105 lines were provided by the USDA National Plant Germplasm System, and 11 were provided by the Chinese Crop Germplasm Resources Information System. Three *Aegilops* accessions were collected from Henan, China by the senior author's lab. Detailed *Aegilops* species information is provided in Supplementary Data 4.

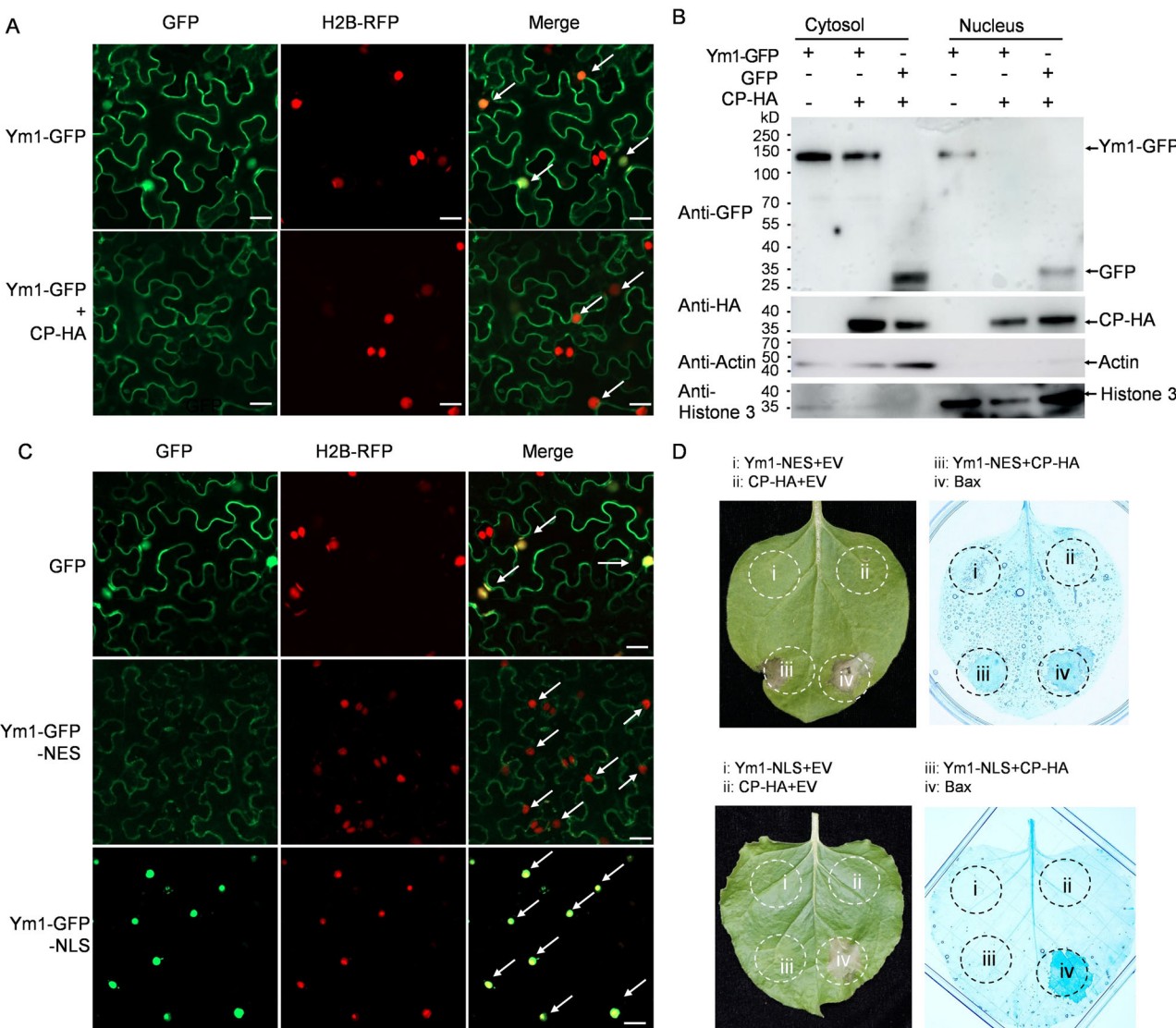

**Fig. 5 | Cytosol localization of CP-Ym1 interaction is essential for cell death triggering activity. A** Ym1-GFP alone was subcellular localized in nucleolus and cytoplasm (upper panel), Ym1-GFP together with WYMV-CP was subcellular localized only in cytoplasm (lower panel). *Agrobacterium* mixture containing plasmids expressing Ym1-GFP and CP-HA fusion proteins were infiltrated into *N. benthamiana* leaves. H2B-RFP was used as the nucleus marker. The signals were observed under a confocal microscope at 2 days post inoculation (dpi). The arrows point the merged signals within nucleus. Bars = 25 μm. The experiment was repeated three times independently with similar results. **B** Ym1-GFP and CP-HA extracted from the cytosol and nucleus components of *N. benthamiana* leaf issues were detected by Western blotting with anti-HA or anti-GFP antibodies. Cytosol-localized protein Actin and nucleus-localized protein Histone 3 were detected with their monoclonal antibodies and used as positive controls of cytosol and nucleus components, respectively. The experiment was repeated three times independently with similar results. **C** Subcellular localization of the Ym1 fused with nucleus exporting (Ym1-GFP-NES) and localization (Ym1-GFP-NLS) sequences. GFP and the fusion proteins, Ym1-GFP-NLS, and Ym1-GFP-NES, were expressed in *N. benthamiana* leaves by Agro-infiltration. The signals were observed under a confocal microscope at 2 dpi. Histone 2B (H2B-RFP) was used as nucleus-localized marker protein. The arrows point the merged signals within nucleus. Bars = 25 μm. The experiment was repeated three times independently with similar results. **D** Analysis of cell death-inducing activity of the Ym1-GFP-NES and NLS fusion proteins. The cell death triggered by each fusion protein was scored at 4 dpi (leaf panel). A thumbnail diagram indicates the positioning of each expressed fusion protein.

## WYMV disease screening in field trials

The above-mentioned plant materials were grown in the WYMV nursery at Zhumadian, Henan Province, China in 2022 and 2023 wheat grown seasons (November to May next year) under natural infection, which is WYMV epidemic annually. The individuals were planted in 1 m-length rows with 15 seedlings per row and 25 cm spacing between rows. For every 10 rows, we grew 1 row of wheat cv. YNXM as the resistant control, 1 row each of wheat cv. YM158 and 2011I-78 as the susceptible controls. Management of the nursery followed standard practices in local wheat production. WYMV disease symptoms of plants in the field nursey were scored three times (Feb 19th and March 10th, 2023, March 2nd, 2024) using a 0–5 rating scale described before[39].

## WYMV resistance evaluation and quantitation of fungal growth in growth chamber

WYMV resistance of transgenic plants and genome edited plants was evaluated in a growth chamber at 8 °C with a 16 h light/8 h dark photoperiod. YM158 was used as susceptible control, Fielder and YNXM were used as resistant controls. Artificial inoculation was performed using the WYMV infectious clones pCB-SP6-R1 and pCB-SP6-R2 following the protocol previously described by Zhang et al.[71]. The WYMV resistance of test materials was determined by the abundance of WYMV CP transcripts in the leaves with qPCR (see below for details) at 14 days after inoculation and disease symptoms. The primer pairs information was listed in Supplementary Data 1.

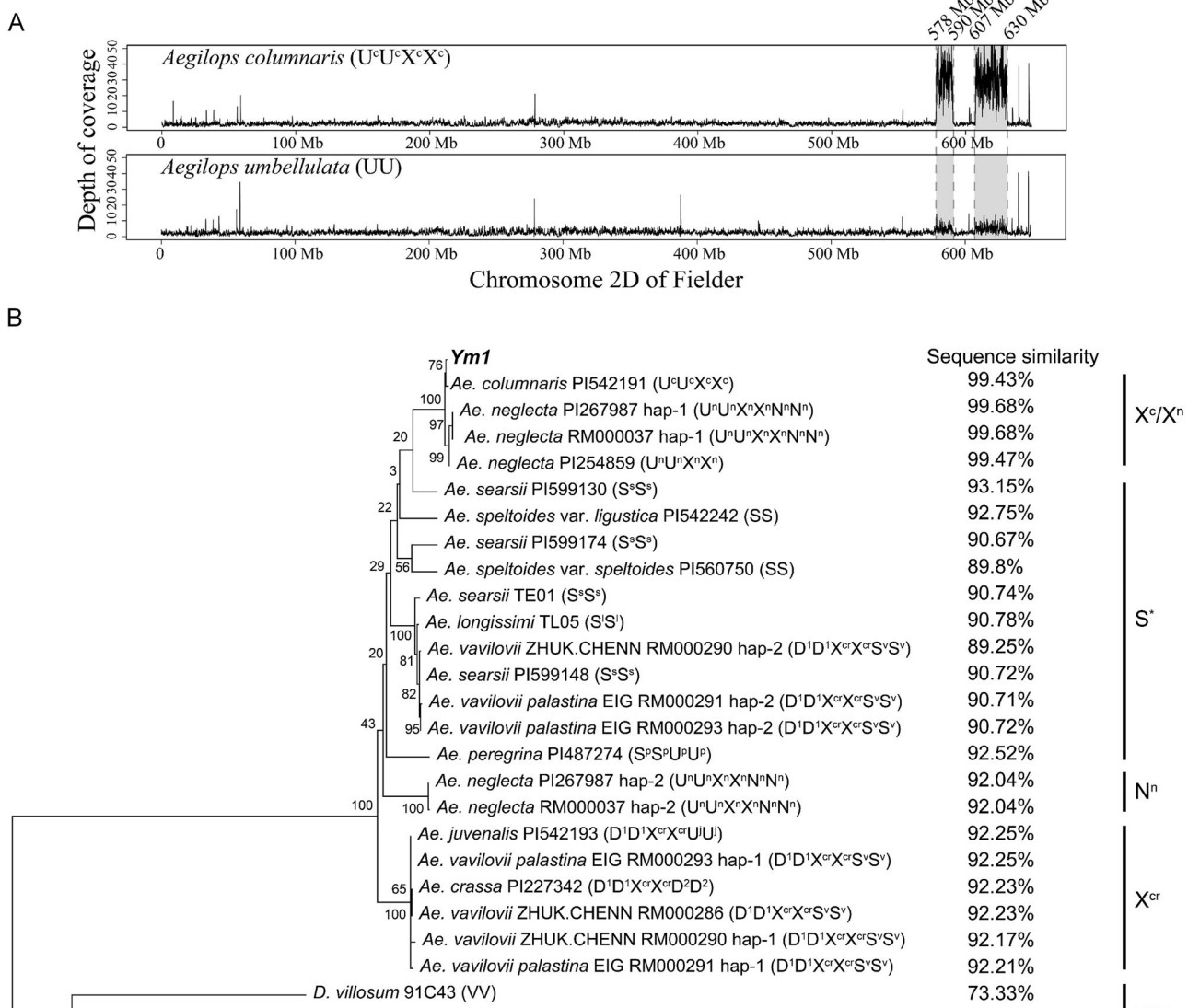

**Fig. 6 | The putative origin of *Ym1*. A** Alignment of resequencing short reads of *Aegilops colummaris* and *Ae. umbellulata* against the 2D chromosome sequence of Fielder. *Y*-axis represents the depth of coverage of mapped reads in a 100 kb sliding window and the *X*-axis is the coordinates of the Fielder 2D chromosome assembly. The grey area indicates the putative introgression region. **B** Phylogram built with sequences of *Ym1* and its orthologs from wheat relatives. The numbers (in percentage) on the right indicate sequence similarities between coding sequences of *Ym1* and its homologs from the investigated species.

The abundances of the vector fungus *Polymyxa graminis* (*Pg*) in the roots of different genotypes and their rhizosphere soils was quantified with qPCR. Test plants were grown in sterile vermiculite under a 14 h light/10 h dark light regime at 24/18 °C and 70% RH. At the 4th-true-leaf stage, seedlings were washed off vermiculite and transplanted into *Pg*-containing soils collected from WYMV nursery. The rhizosphere soil and root samples were collected at 14 d after transplanting for DNA extraction using the DNeasy PowerSoil kit (Qiagen, Germany). The abundance of *Pg* was determined with qPCR as described by Wu et al.[72]. The primer pairs information was listed in Supplementary Data 1.

### RT-PCR and qPCR

Total RNA was isolated using the TRI-Zol reagent (Invitrogen, USA). The HiScript® II First Strand cDNA Synthesis Kit (Vazyme, Nanjing, China) was used to synthesize the first-strand cDNA with 1 μg total RNA. Each 20 μL reaction contains 10 μL 2 × RT Mix, 4 μL HiScript II Enzyme Mix, 1 μL oligo (dT)23VN (50 μM), and nuclease-free $H_2O$ up to 20 μL. qPCR was performed on a LightCycler® 480 instrument (Roche, Germany) using SYBR Green qRT-PCR mixture (Vazyme Biotech, Nanjing, China) with the following conditions: 95 °C for 30 s, 40 cycles at 95 °C for 5 s, and 60 °C for 30 s, a dissociation step at 95 °C for 15 s, 60 °C for 60 s. The wheat *Tubulin* gene was used as an internal control, and the fold change was calculated using the $2^{-\Delta\Delta CT}$ method. RT-PCR was used to detect the expression of *WYMV CP*[39] and *Ym1*. All the primers used in this study are listed in Supplementary Data 1.

### Molecular markers, gene annotation, and *Ym1* sequence analysis

For fine mapping of the resistance QTL, SSR, Indel, and SNP markers were developed as described previously[39]. The protein-coding sequences of 57 annotated genes located in the target region of the Fielder reference genome were submitted to the eggnog website (http://eggnog5.embl.de/#/app/home) for manual annotation[73]. The full-length *Ym1* CDS was amplified by RT-PCR from YNXM and Fielder

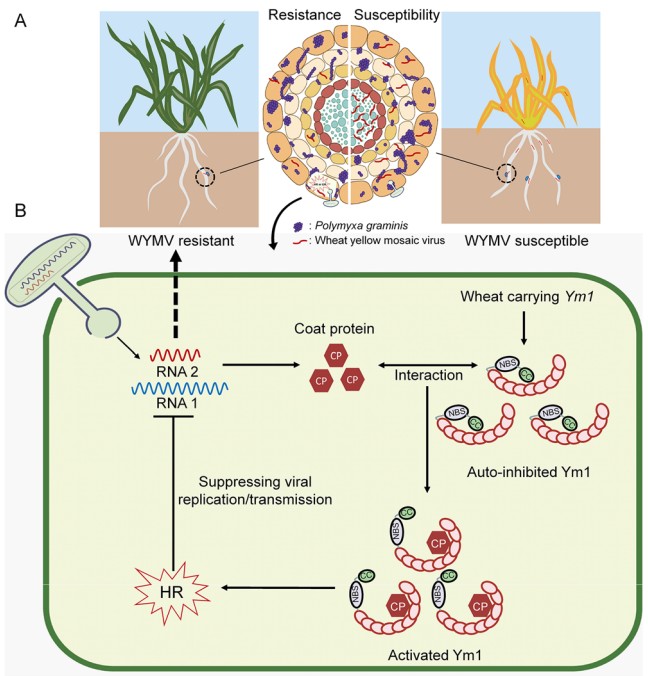

**Fig. 7 | A working model of *Ym1*-mediated wheat yellow mosaic virus (WYMV) resistance. A** *Ym1* confers WYMV resistance by blocking virus invasion other than *Polymyxa graminis* (*Pg*) fungal colonization in the roots. In the left, resistant wheat plant carrying *Ym1* and its transverse root; In the right, susceptible plant without *Ym1* and its transverse root. In the transverse root model from outer to inner are the epidermis, cortex, endodermis, pericycle and stele cells. The purple elements indicate the colonized *Pg*; the red curve lines indicate the infected virus. **B** Diagram showing the Ym1-coat protein (CP) recognition system in wheat resistance against WYMV. *Ym1* is root specifically expressed and upregulated upon viral infection. Without viral infection, Ym1 is present as auto-inhibited status. When the virus vectored by *Pg* infects wheat root cells, auto-inhibited Ym1 is activated by recognizing and interacting with WYMV CP. The activated Ym1 triggers hypersensitive response (HR) or extreme resistance (ER) to inhibit viral replication or movement, results in ultimate outcome to block the virus invasion into root stele.

using primer pair Ym1-F/R followed by Sanger sequencing. The confirmed *Ym1* sequence was further analyzed using ORF Finder software (https://www.ncbi.nlm.nih.gov/orffinder/). The conserved domains in the *Ym1* sequence were predicted at the InterPro website (https://www.ebi.ac.uk/interpro/)[74]. The ORF sequences of *Ym1* homologs were cloned from the genome DNA of *Aegilops* accessions using primer pair Ym1-F/R.

### Transcriptome analysis

Root and leaf tissues of YNXM grown in WYMV nursery and WYMV-free field (three biological replications each) were collected for RNA-Seq with the Illumina HiSeq 2500 platform. High-quality sequencing reads were mapped onto the Fielder reference genome[75] using HISAT2 v2.2.1[76]. The transcripts per kilobase of exon model per million mapped reads (TPM) of each gene were calculated based on the length of the gene and the read count mapped to this gene to reflect gene expression levels. DEG analysis was performed using the DESeq2 v4.3.3 R package between inoculated and non-inoculated control from the same tissue. The resulting *P*-values were adjusted using the Benjamini and Hochberg approach for controlling the false discovery rate. DEGs were identified with thresholds of adjusted $P < 0.05$ and $|Log_2(fold change)| \geq 1$.

### Ethyl methane sulfonate (EMS) mutagenesis of in WYMV resistant YNXM

EMS mutagenesis followed Chen et al.[77]. Specifically, ~5000 YNXM seeds were soaked in 0.05 M phosphate-buffered saline (pH 7.0) for 8 h, then in 0.6% EMS for 16 h in darkness. After washing in running water for 2 h, the $M_0$ seeds were germinated and planted in the field. A total of 1803 $M_1$ and derived $M_2$ lines were evaluated for WYMV resistance in two WYMV nurseries: Zhumadian of Henan Province in two growing seasons (2018–2019 and 2020–2021), and Yandu of Jiangsu Province in one growing season (2020–2021). The *CP* abundance was analyzed by qRT-PCR. The primer pairs information was listed in Supplementary Data 1.

Three stable independent WYMV-susceptible mutant lines were identified. The YNXM and three mutant lines were subjected for BGISEQ-500 paired-end shotgun sequencing at BGI-Shenzhen, China following the Whole Genome Resequencing PCR-free Library Preparation Pipeline (DNBSEQ; BGI-NGS-JK-DNA-003 A0). After initial quality control filtration, the clean reads were mapped onto the Fielder reference genome sequence using BWA-mem (v0.7.17-r1188)[78]. The SNPs and insertion/deletion (InDels) were called using the GATK pipeline[79], and the filtered SNPs were annotated by snpEFF[80].

### Virus-induced gene silencing (VIGS) in WYMV resistant YNXM

VIGS procedure followed Yuan et al.[81]. The Ym1 CC (228 - 398 bp in CDS) and LRR (2034 - 2218 bp in CDS) domains were targeted for VIGS. The cDNA sequences were amplified with the primer pairs, BSMV-γ-Ym1-CC-rcseq1-F/R and BSMV-γ-Ym1-LRR-rcseq1-F/R, and inserted into the *Apa* I site of vector pCaBS-γ to produce pCaBS-γ:Ym1$^{CC}$ and pCaBS-γ:Ym1$^{LRR}$, respectively. The *Agrobacterium* strain EHA105 culture harboring pCaBS-γ recombinant vectors was mixed with the EHA105 culture harboring pCaBS-α and pCaBS-β in 1:1 ratio. The mixture was infiltrated into *N. benthamiana* leaves. The recombinant vectors pCaBS-γ:TaPDS and pCaBS-γ were used as the positive and negative controls, respectively. Seven days after infiltration, 1 g leaf tissues showing BSMV infection symptoms were collected and ground in 3 ml BSMV inoculation buffer (5.6 mM $H_2NaPO_4$, 14.4 mM $HNa_2PO_4$, 10 g/L celite). The resulting solution was used to inoculate two-week-old wheat seedlings in the WYMV nursery. Ten days after inoculation, BSMV infection symptoms were observed on new leaves of all BSMV-inoculated seedlings, and photobleached leaves were observed on the BSMV-TaPDS inoculated ones. The gene expression level of *Ym1* in the roots of inoculated seedlings was examined using qPCR. The WYMV resistance was determined by phenotypes on inoculated leaves and RT-PCR with WYMV CP specific primers. The CP abundance was analyzed by qRT-PCR. The primer pairs information was listed in Supplementary Data 1.

### CRISPR-Cas9-based genome editing of *Ym1* in WYMV resistant Fielder

For targeted editing of *Ym1* with CRISPR-Cas9, the WYMV-resistant Fielder was transformed with *Agrobacterium* strain EHA105 harboring two vectors of pBUE411-Ym1, which targeted Ym1 CC and LRR domains. The target site sequences are 5'-gattcatgaagaagacccgCGG-3' and 5'-caaacttcgcctccgaactAGG-3', respectively. Positive transgenic seedlings were selected by PCR using the primers designed on the Cas9 coding sequence. Two CRISPR-Cas9 target sites in the positive plants were amplified and genotyped by Sanger sequencing of multiple colonies harboring cloning vectors with PCR fragments. Long fragment deletion mutants generated by simultaneous double-strand break-induced on two target sites were also screened using PCR flanking both target sites, and short PCR fragments were sequenced by Sanger sequencing to confirm the genotypes. The CP abundance and *Ym1* expression were analyzed by qRT-PCR. The primer pairs information was listed in Supplementary Data 1.

### Genetic transformation of *Ym1* in WYMV susceptible YM158

The CDS of *Ym1* was cloned and inserted into the expression vector pWMB110 with the *Ubiquitin* promoter between two restriction endonuclease sites *Bam*HI and *Sac*I. The overexpression vector

pWMB110Ubi:*Ym1* was introduced into immature embryos of susceptible wheat variety YM158 by *Agrobacterium tumefaciens*-mediated transformation through a commercial service[82]. Specific primer pair Ym1-DM-F/R was designed to screen positive transgenic plants in T$_0$ plants. The *Ym1* expression level in the leaves of transgenic plants was analyzed and normalized to the expression of the wheat *Tubulin* gene by qPCR. The CP abundance and *Ym1* expression were analyzed by qRT-PCR. The primer pairs information was listed in Supplementary Data 1.

### RNA in situ hybridization

Tissue fixation and RNA in situ hybridization were performed as described previously[83] with minor modifications. In brief, root and shoot tissues were chopped before infiltration to facilitate fixative penetration. The 879-bp (4 to 882 nt) cDNA sequence of WYMV CP and 415-bp (2156 to 2570 nt) *Ym1* fragment at 3′ terminal (Supplementary Data 1 and Supplementary Data 6) were used as the probes. The amplicons were inserted into the pGM-T vector (TIANGEN, China), which was subsequently linearized by either *Apa*I (to use SP6 promoter) or *Pst*I (T7 promoter) digestion. SP6 RNA polymerase was used to generate an antisense probe and T7 RNA polymerase was used to generate a sense probe. Probes for in situ hybridization were transcribed using the Dig RNA labeling kit (Roche).

### WYMV transmission experiment

*Agrobacterium* harboring WYMV infectious clones mixed with that harboring Ym1-GFP constructs were co-infiltrated into *N. benthamiana* leaves for 3d under 25 °C. The GFP empty vector and *Agrobacterium* mock were used as negative controls. The leaves were cut and smashed onto the PVDF membrane, and then analyzed by Western blot using antibody anti-CP (1:5000, kindly provided by Professor Jiang Yang, Ningbo University).

### Cell death assay in *N. benthamiana* leaves

The full-length CDS of 11 WYMV protein coding genes, *Ym1* without stop codon and its truncated domains: CC, CC-NES, CC-NLS, NBS, LRR, CC-NBS, NBS-LRR (without stop codon) were individually inserted into the vector pBin-HA, and then introduced into *Agrobacterium tumefaciens* strain GV3101 for the infiltration into *N. benthamiana* leaves. Agrobacteria suspensions were infiltrated at OD$_{600}$ = 0.8. The cell death was observed at 48–72 h after infiltration. The leaves were then stained with trypan blue and subsequently de-stained in 2.5 g ml$^{-1}$ chloral hydrate in distilled water. The staining reactions were observed under white light channel of a microscope[84].

### Cell death assay in wheat roots

WYMV infectious clones were artificially inoculated into roots wheat plants grown in a growth chamber for 5 d at 8 °C with a 16 h light/8 h dark photoperiod. The roots of the seedlings were submerged in PI (10 mg/ml) and images were immediately captured. The live and dead cells were visualized and images were captured under a confocal microscope (Carl Zeiss, Jena, Germany).

### Yeast two-hybrid (Y2H) assay

For Y2H, the full-length CDS of WYMV including P1, 7 K, 14 K, CI, NIa-pro, VPg, NIb, CP, P2, P3, and P3N-PIPO was individually inserted into the pGADT7 vector. The full-length CDS of *Ym1* without stop codon and its CC, NBS, and LRR (without stop codon) domains were respectively inserted into pGADT7 and pGBKT7 vectors. The *Ym1* alleles from three mutants and LRR domain from YNXM-Mut3 were inserted into pGBKT7 vectors. Yeast solution (strain AH109) with recombinant pGADT7 and pGBKT7 combinations were plated onto a low-stringency selective medium lacking tryptophan and leucine (SD/-Trp-Leu) for 3 days at 30 °C and then plated onto a high-stringency selective medium lacking tryptophan, leucine, histidine, and adenine (SD/-Trp-Leu-His-Ade). Yeast cells carrying pGBKT7-53 or pGBKT7-Lam were used as positive or negative controls, respectively.

### Firefly luciferase complementation imaging (LCI) assay

For LCI assay, the coding sequences of WYMV *CP* and wheat *Ym1* with its truncated LRR domain were amplified and inserted into pCambia1300-nLuc and pCambia1300-cLuc vectors to produce CP-nluc, cluc-Ym1, and cluc-Ym1$^{LRR}$, respectively. *Agrobacterium* cells carrying nluc-CP and an equal amount of *Agrobacterium* cells carrying cluc-Ym1 and cluc-Ym1$^{LRR}$ were mixed and infiltrated into *N. benthamiana* leaves. After 2 d, the same leaves were infiltrated again with a 0.2 mM luciferin (Thermo Scientific, USA) solution followed by the detection of luciferase activity via a low-light cooled charge-coupled device imaging apparatus under a microscope.

### Co-immunoprecipitation (co-IP) and western blotting assays

Ym1 and LRR were cloned into the pCambia1305.1-GFP vector. The WYMV CP was cloned into the pBin-HA vector. The constructs were transformed into *Agrobacterium* strain GV3101 and infiltrated into the abaxial surface of 4–6-week-old *N. benthamiana* leaves. For co-IP, the *N. benthamiana* leaves were sampled at 48 h after agroinfiltration. Total proteins were extracted using a native extraction buffer (50 mM Tris-MES, pH 8.0, 0.5 M sucrose, 1 mM MgCl$_2$, 10 mM EDTA, 5 mM DTT, 1 mM phenylmethyl sulfonyl fluoride, and protease inhibitor cocktail). The supernatant of each sample was incubated with Anti-GFP Mag-Beads (Yeasen, 20564ES76) at 4 °C for 2.5 h with gentle agitation. The beads were collected on a magnetic rack and washed five times using a native extraction buffer. The bound protein complexes were eluted by boiling in the SDS loading buffer for 15 min and analyzed by SDS-PAGE and immunoblotting using anti-HA (Abmart, M20003) or anti-GFP (Abmart, M20004).

### Subcellular localization

The full-length CDS of *Ym1* was inserted into the vector pCambia1305.1-GFP for cellular translocation. The nuclear localization signal (NLS, QPKKKRKVGG) or nuclear export signal (NES, NELALKLAGL-DINK) were fused to the C-terminus of *Ym1-GFP*. Those resulted plasmids were transformed into the *Agrobacterium* strain GV3010 and infiltrated into *N. benthamiana* leaves. At 48 h after infiltration, the locations of the above fusion proteins were examined under a confocal microscope (Carl Zeiss, Jena, Germany). The excitation wavelengths of GFP and mRFP were 488 nm and 561 nm, and the emission wavelengths were 495–530 nm and 600–650 nm, respectively. The cytoplasmic and nuclear fractionation followed an early study[46]. The fraction protein was analyzed by SDS-PAGE and immunoblotting using anti-HA (Abmart, M20003) or anti-GFP (Abmart, M20004) antibodies. Anti-histone H3 and anti-Actin antibodies were used as nuclear and cytoplasmic localized markers, respectively.

### Phylogenetic analysis

The phylogenetic tree of *Ym1* and homologs was built using a Maximum Likelihood Estimate implanted in MEGA-X[85] and visualized on iTOL[86]. The polypeptide sequences of NBS-LRR type disease resistance proteins from *Triticum* spp. and related genera were aligned using Muscle v5.1 software. The sequences were assembled to form a maximum likelihood phylogenetic tree tested by bootstrapping (1000 replications) using FastTree v2.1.11 software[87]. Phylogenetic trees were processed and visualized using FigTree and iTOL.

### Sequence alignment and microsynteny analysis

To reveal the putative original alien species of *Ym1*, one hundred million reads were randomly selected from the clean reads of *Ae. columaris* and *Ae. umbellulata*, and mapped to the Fielder reference genome sequence using BWA-mem (v0.7.17-r1188)[78]. The averaged

coverage of mapped reads in each 100 Kb sliding window was calculated using SAMtools (v0.1.19-44428 cd)[88].

Microsynteny of 57 genes within the 5.6 Mbp *Ym1* regions of Fielder and other Triticeae species was examined using the module 'jcvi.compara.synteny' of MCscan (Python version[89] with the '--iter=0.99' setting. Details of the 57 genes are listed in Supplementary Data 2.

## Reporting summary
Further information on research design is available in the Nature Portfolio Reporting Summary linked to this article.

## Data availability
The nucleotide sequences of *Ym1* and its coding protein were obtained from the GenBank database: PP909815. The transcriptomic and resequencing data generated in this study have been deposited in the Genome Sequence Archive (Genomics, Proteomics & Bioinformatics 2021) in National Genomics Data Center (Nucleic Acids Res 2022), China National Center for Bioinformation/Beijing Institute of Genomics, Chinese Academy of Sciences under accession code GSA: CRA023357, CRA023440 that are publicly accessible at https://ngdc.cncb.ac.cn/gsa. Phenotype data generated or analyzed during this study is included in this published article (and its supplementary files). Genotype data are available from the corresponding author upon reasonable request. The source data underlying Figs. 1–5, as well as Supplementary Fig. 5 and 6 are provided as a Source Data file. Source data are provided with this paper.

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

## Acknowledgements

This work was supported by the National Natural Science Foundation of China (No. 32472142), Fundamental Research Funds for the Central University (NO. XUEKEN2022012), Jiangsu Provincial Key Research and Development Program (BE2022346), the Key Research and Development Major Project of Ningxia Autonomous Region (No. 2019BBF02022-04), the Jiangsu Agricultural Technology System (JATS) (NO. JATS [2023] 422), Seed Industry Revitalization Project of Jiangsu Province (JBGS (2021)006), the Joint Research of Wheat Variety Improvement of Anhui (2021-). We thank Professors Yiqun Weng (University of Wisconsin-Madison, USA), Jianping Chen (Ningbo University, China), Xueyong Zhang (Chinese Academy of Agricultural Sciences, China), Feng Chen (Henan Agricultural University, China), Ping Yang (Chinese Academy of Agricultural Sciences, China), Robert McIntosh (University of Sydney, Australia) and Evans Lagudah (CSIRO, Australia) for their valuable comments, critical proofreading and improvement of our manuscript.

## Author contributions

Conceptualization, X.W. and J.X.; Methodology, Y.C., D.K., H.W., X. Tao, and J.Y.; Investigation, Y.C., D.K., J.J., L.W., X. Tang, K.D., M.W., W.C., J.L., and X.F.; Formal Analysis, Y.C., D.K., J.X., and C.J.; Visualization, Y.C., X.Z., L.S., and H.Z.; Writing - Original Draft, Y.C., D.K., J.X., and X.W.; Writing - Review & Editing, X.W., J.X., and H.W.; Funding Acquisition, X.W. and J.X.; Resources, X.W., Z.W., M.W., H.G., and H.W.; Supervision, X.W., J.X., B.S., W.W., and H. W.

## Competing interests

A patent on the application of *Ym1* in wheat breeding was filed by Xiue Wang, Haiyan Wang, Yiming Chen, Dehui Kong, Jin Xiao, Chunxia Yuan. The remaining authors declare no competing interests.

## Ethical approval

On behalf of all authors, the corresponding author states that the experiments comply with the current laws of the country in which they were performed. All the authors declare that they are all engaged in the accomplishment of the work.
