## [Transparent Peer Review file · Nature Communications]

A wheat CC-NBS-LRR protein TaRx confers WYMV resistance by recognizing viral coat protein

Corresponding Author: Professor Xiue Wang

Version 0:

Reviewer comments:

Reviewer #1

(Remarks to the Author)

In this manuscript, the authors separated the TaRx gene by map-based cloning from QYm.nau-2D locus, a most widely used for wheat yellow mosaic virus (WYMV) control. TaRx is a typical CC-NBS-LRR type R protein. It is specifically expressed in root and induced upon WYMV infection. TaRx confers WYMV resistance by blocking viral transmission from the root cortex into steles and further movement to aboveground tissues. Moreover, TaRx CC domain can trigger cell death in tobacco plants. TaRx physically interacts with the WYMV coat protein (CP), leading to nucleocytoplasmic redistribution. These findings are interesting in appearance, however, there are high gaps of the functions of TaRx with the viral resistance. This is because virus is a cellular pathogen whose invasion into the host plant is dependent on the vector. Viral movement in host plant is associated with the vascular tissue. Although TaRx can caused cell death and have nucleocytoplasmic redistribution by interacting with CP in tobacco cells, what is the effects of TaRx on the virus invasion or movement in wheat plants. These functions of TaRx, found by the authors, is similarly to the other R protein, such as MLA that confer the resistance to the fungus. In addition, this gene is widely used in the breeding of common wheat, thus weakening the potentially applicated merit of TaRx in the future.

Major points:

1. As stated by the authors, TaRx confers WYMV through blocking virus invasion into the root of host plant. What is the molecular mechanism of TaRx in hindering viral invasion. If this is because of the cell death triggered by TaRx and CP interaction, the author should demonstrate and observe the cell death in root cells of virus-infected wheat.
2. What is the relationship on the molecular level between hypersensitive response (HR) caused by TaRx and WYMV resistance of TaRx. What are the effects of HR on the Pg colonization and infection.
3. TaRx is specially induced to expressed in root of WYMV-infected wheat. What is the orthologs of TaRx (hap-1 and hap-2) in expression. The expression of TaRx orthologs is associated with their resistance to WYMV?
4. The alleles of TaRx in three YNXM mutants (Figure 1) can interact with WYMV CP? These interactions can cause cell death or HR in plant cells?
5. The interaction of LRR domain with CP is special? LRR domains in other R protein, such as Yr10 can interact with CP?
6. TaRx can function in the movement of WYMV in wheat plant?

Reviewer #2

(Remarks to the Author)

Wheat yellow mosaic virus (WYMV) is a significant soil-borne fungal virus that severely threatens wheat production and food security in East Asia. Currently, identifying disease resistance genes is the most environmentally friendly and effective approach for controlling viral diseases. In particular, dominant NLR genes, which can confer complete immunity to viruses, hold great value in crop breeding. Although the CNL-type gene Ym2 that confers resistance to WYMV has been reported (Mishina et al., 2023, PNAS), further identification of resistance NLR genes will provide additional genetic resources for improving crop resistance to WYMV.

In this work, the authors cloned the WYMV-resistant gene TaRx from QYm.nau-2D in wheat using positional cloning techniques. This gene is a CNL-type R gene, and the authors genetically demonstrated through gene editing and overexpression that TaRx confers resistance to WYMV in wheat from the root, aligning with WYMV's transmission pathway. Additionally, interaction assays showed that the LRR domain of TaRx can specifically recognize WYMV's CP, triggering cell

death. This recognition process primarily occurs in the cytoplasm. The authors also conducted an evolutionary analysis of TaRx. Overall, the manuscript is well-written and easy to follow, and the data support the main conclusions of this work. I have some suggestions for the authors' consideration.

1. I suggest renaming TaRx, as there are already resistance genes named Rx and Rx2 for Potato virus X. Naming another anti-WYMV R gene as Rx might cause confusion.
2. Line 526: Since WYMV infection requires low temperatures (8°C treatment), it would be interesting to test whether TaRx activity is also temperature-dependent. The authors could examine whether the cell death induced by co-infiltrating TaRx and CP is affected by temperature.
3. Figure 1 and 4F: (1) The authors obtained several TaRx mutants (YNXM-Mut1-3) through EMS mutagenesis. I suggest including these mutants as controls in the cell death assay shown in Figure 4F; (2) In Figure 4F, both TaRx and CP are tagged with GFP. The addition of relatively large tags can potentially affect NLR function. I wonder how the results would compare if smaller tags, like HA or FLAG, were used with both TaRx and CP.
4. Figure 5E: The authors only used the CC domain to observe the impact of NLS and NES fusion on cell death, which may not fully reflect changes in TaRx activity. I suggest using full-length TaRx-NES and TaRx-NLS co-injected with CP to observe cell death. Additionally, as the authors have an infectious WYMV clone, I wonder if WYMV can infect *N. benthamiana*. If so, I also suggest testing the anti-WYMV activity of different TaRx derivatives, such as YNXM-Mut1-3 and TaRx-NES/NLS.

Reviewer #3

(Remarks to the Author)

In this study, Chen et al. identified a wheat R gene named TaRx, which confers resistance to wheat yellow mosaic virus (WYMV). The authors describe TaRx as a typical CC-NLR protein that recognizes the coat protein of WYMV. Moreover, they suggest that the CC domain of TaRx functions as the cell death executor and that its cytoplasmic localization is involved in the cell death response. The discovery of the wheat R gene is important for global food security, and mechanistic insights into TaRx function are of broad interest within the plant biology research field. The quality of the data, data analyses, and figure descriptions—especially in the genetics section—are good and effectively support the authors' conclusions. I have a few comments on the functional analysis section. I have listed some comments that I hope will be useful for improving the manuscript:

Major comments:

1. The name TaRx may be confusing, as researchers often refer to StRx from potato as Rx. Additionally, TaRx and StRx are not orthologous genes. I suggest the authors reconsider the gene name.
2. Line 285,286: The authors concluded that WYMV CP interacts weakly with full-length TaRx and strongly with the TaRx LRR domain. This difference may be due to varying protein expression levels in yeast. To support Figure 4B, the authors should perform western blot analyses to check the accumulation levels of full-length TaRx and each of its domains.
3. Figure 5C-E: Testing the function of NES- and NLS-fused proteins is interesting; however, functional validation of NES- and NLS-fused full-length TaRx would be more informative than testing just the CC domain. I suggest that the authors assess the expression of TaRx-NES or TaRx-NLS with CP in both the cell death assay and cell biology experiment.

Minor comments:

1. Figure description of Figure 3D, E: I suggest clarifying which genes (TaRx or CP) are being detected in these figures.
2. Line 288: I do not think that Co-IP and LCI assays detect the "physical interactions" between two proteins, as endogenous proteins in plants could mediate the association between them. The terms "interaction" or "association" would be more appropriate here.
3. Figure 4F: It would be helpful to mention in the manuscript that TaRx/CP-induced cell death is weaker compared to CC and Bax, which is why trypan blue staining was conducted.
4. Line 403-404: the authors should avoid mentioning the content of "data not shown."
5. Figure legend of Figure 4 is not consistent with the current figure. Please revise this.

Version 1:

Reviewer comments:

Reviewer #1

(Remarks to the Author)

Authors have revised the manuscript according to my comments, I agree to accept it now.

Reviewer #2

(Remarks to the Author)

The author has addressed my concerns.

Reviewer #3

(Remarks to the Author)

The authors addressed all of my concerns and suggestions. I do not have any other comments to the revised manuscript.

Response to Reviewers

Reviewer #1 (Remarks to the Author):

In this manuscript, the authors separated the TaRX gene by map-based cloning from QYm.nau-2D locus, a most widely used for wheat yellow mosaic virus (WYMV) control. TaRx is a typical CC-NBS-LRR type R protein. It is specifically expressed in root and induced upon WYMV infection. TaRx confers WYMV resistance by blocking viral transmission from the root cortex into steles and further movement to aboveground tissues. Moreover, TaRx CC domain can trigger cell death in tobacco plants. TaRx physically interacts with the WYMV coat protein (CP), leading to nucleocytoplasmic redistribution. These findings are interesting in appearance, however, there are high gaps of the functions of TaRx with the viral resistance. This is because virus is a cellular pathogen whose invasion into the host plant is dependent on the vector. Viral movement in host plant is associated with the vascular tissue. Although TaRx can caused cell death and have nucleocytoplasmic redistribution by interacting with CP in tobacco cells, what is the effects of TaRx on the virus invasion or movement in wheat plants. These functions of TaRx, found by the authors, is similarly to the other R protein, such as MLA that confer the resistance to the fungus. In addition, this gene is widely used in the breeding of common wheat, thus weakening the potentially applied merit of TaRx in the future.

Major points:

Point 1: As stated by the authors, TaRx confers WYMV through blocking virus invasion into the root of host plant. What is the molecular mechanism of TaRx in hindering viral invasion. If this is because of the cell death triggered by TaRx and CP interaction, the author should demonstrate and observe the cell death in root cells of virus-infected wheat.

Response: Our results showed TaRx (According the Reviewers' suggestion, we renamed TaRx as Ym1 in the revised manuscript) and CP interaction triggered cell death in tobacco leave cells. To observe whether this also happens in root cells of wheat infected by WYMV, we artificially inoculated WYYV infectious clones into roots of wheat genotypes having or having not the Ym1. Cell death was only detected in roots of genotypes having *Ym1* gene (Fielder and *Ym1* transgenic plant), while not in roots of genotypes not having *Ym1* gene (*Ym1* knockout line CRTa-4-2 and Yangmai 158). These demonstrated that Ym1 could hinder viral invasion by recognition of CP by Ym1, which led to the elicitation of cell death.

We added the results in Figure 4G and described the results in lines 303-307 of the revised manuscript.

Point 2: What is the relationship on the molecular level between hypersensitive response (HR) caused by TaRx and WYMV resistance of TaRx. What are the effects of HR on the *Pg* colonization and infection.

Response: As illustrated in our response to point 1, we only observed HR (cell death) in roots of genotypes having *Ym1*, including Fielder and *Ym1* transgenic lines overexpressing *Ym1* gene. We also conducted cell death assay using mutated *Ym1* alleles in the three susceptible EMS-induced mutants. The co-infiltration of mutated *Ym1* with CP failed to elicit cell death. Thus, we assumed that HR was associated with Ym1-conferred WYMV resistance. We added the results in Figure 4G and described the results in lines 300-304 of the revised manuscript.

We also provided the fungal abundance measurement results by artificially inoculated *Pg* fungi. No significant difference was detected among the genotypes having or not having *Ym1* gene (Figure

2; Figure S6). The *Ym1* does not contribute to the preventing of fungal colonization and infection in the roots (please refer to the results in lines 254-262). Therefore, we propose HR elicited by *Ym1* has no effect on *Pg* colonization and infection.

Point 3: TaRx is specially induced to expressed in root of WYMV-infected wheat. What is the orthologs of TaRx (hap-1 and hap-2) in expression. The expression of TaRx orthologs is associated with their resistance to WYMV?

Response: This is a very good point worthy to study. We investigated the gene expression of *Ym1* orthologs in roots of wild species including *Ae. columnaris*, *Ae. neglecta* and *Ae. vavilovii* in response to artificial inoculation of WYMV infectious clones. The expression of all the tested ortholog genes were induced by WYMV infection in roots, while CP transcript was undetectable. This is consistent with our assumption that *Ym1* is an introgression from *Aegilops* species, and the *Ym1* orthologs also contribute to WYMV resistance.

We added the results in Supplementary Figure 13 and described them in lines 377-383 of the revised manuscript.

Point 4: The alleles of TaRx in three YNXM mutants (Figure 1) can interact with WYMV CP? These interactions can cause cell death or HR in plant cells?

Response: Thanks for the comments. We performed Y2H to test whether the mutation of *Ym1* affect the interaction between WYMV and CP. The mutated *Ym1* allele in YNXM-Mut2 failed to interact with CP due to the truncation in its LRR domain (Figure S9). Co-infiltration of mutated *Ym1* allele in YNXM-Mut2 and CP could not elicit cell death (Figure S9). The mutated *Ym1* alleles in YNXM-Mut1 and -Mut3 interacted with CP, however neither of the interaction could elicit cell death. These results could explain why the three mutants are all WYMV susceptible.

We added the results in Supplementary Figure 9 and described them in lines 308-314 of the revised manuscript.

Point 5: The interaction of LRR domain with CP is special? LRR domains in other R protein, such as Yr10 can interact with CP?

Response: Thanks for the suggestion. We tested the interaction of the LRR domains of four previously reported R proteins (i.e. *Ym2*, *StRx1*, *Mla10*, *Yr10*) with WYMV CP. None of them interacted, indicating the specificity of the interaction between *Ym1* LRR domain with WYMV CP.

We showed the results in the following **Figure** without adding it in the revised manuscript.

Figure Interaction analysis between WYMV CP and each of the LRR domains from four previously reported R proteins (including Ym2, StRx1, Mla10, Yr10) by yeast two-hybrid (Y2H) assay

Point 6: TaRx can function in the movement of WYMV in wheat plant?

Response: Thanks for the comments. We performed WYMV transmission experiments to test whether Ym1 can function in the movement of WYMV, and the results were shown in Figure S7. When infiltrating the WYMV infectious clones into *N. benthamiana* leaves, the WYMV CP (detected by anti-CP antibody) could be detected surrounding the infiltration sites. While when co-infiltrating Ym1-GFP constructs and WYMV infectious clones into *N. benthamiana* leaves, the CP abundance was obviously reduced and CP distribution was restricted within the infiltration sites. This indicated that Ym1 can function in inhibiting the movement of WYMV.

We added the results in Supplementary Figure 7 and described them in lines 273-277 of the revised manuscript.

Reviewer #2 (Remarks to the Author):

Wheat yellow mosaic virus (WYMV) is a significant soil-borne fungal virus that severely threatens wheat production and food security in East Asia. Currently, identifying disease resistance genes is the most environmentally friendly and effective approach for controlling viral diseases. In particular, dominant NLR genes, which can confer complete immunity to viruses, hold great value in crop breeding. Although the CNL-type gene Ym2 that confers resistance to WYMV has been reported (Mishina et al., 2023, PNAS), further identification of resistance NLR genes will provide additional genetic resources for improving crop resistance to WYMV.

In this work, the authors cloned the WYMV-resistant gene TaRx from QYm.nau-2D in wheat using positional cloning techniques. This gene is a CNL-type R gene, and the authors genetically demonstrated through gene editing and overexpression that TaRx confers resistance to WYMV in wheat from the root, aligning with WYMV's transmission pathway. Additionally, interaction assays showed that the LRR domain of TaRx can specifically recognize WYMV's CP, triggering cell death. This recognition process primarily occurs in the cytoplasm. The authors also conducted an evolutionary analysis of TaRx. Overall, the manuscript is well-written and easy to follow, and the data support the main conclusions of this work. I have some suggestions for the authors' consideration.

Point 1: I suggest renaming TaRx, as there are already resistance genes named Rx and Rx2 for Potato virus X. Naming another anti-WYMV R gene as Rx might cause confusion.

Response: Thanks for the suggestion. We renamed TaRx as Ym1 (Wheat yellow mosaic virus resistance gene 1), and we changed TaRx1 to Ym1 throughout the revised manuscript, Tables and Figures, supplementary materials and Responses to Editor and Reviewers.

Point 2: Line 526: Since WYMV infection requires low temperatures (8°C treatment), it would be interesting to test whether TaRx activity is also temperature-dependent. The authors could examine whether the cell death induced by co-infiltrating TaRx and CP is affected by temperature.

Response: Thanks for the suggestion. We performed the co-infiltration experiment and then put the co-infiltrated *N. benthamiana* plants in a growth chamber at 8°C, 16°C and 25°C for 4 days. The cell death was observed under 16°C and 25°C growth condition rather than that under 8°C growth condition. It indicated that Ym1 activity is not low temperature dependent.

We showed the results in the following Figure without adding it in the revised manuscript.

Figure Observation of cell death symptom elicited by co-infiltrating Ym1 and CP constructs in *N. benthamiana* leaves which is grown under different temperature (8°C, 16°C and 25°C) for 4 days

Point 3: Figure 1 and 4F: (1) The authors obtained several TaRx mutants (YNXM-Mut1-3) through

EMS mutagenesis. I suggest including these mutants as controls in the cell death assay shown in Figure 4F; (2) In Figure 4F, both TaRx and CP are tagged with GFP. The addition of relatively large tags can potentially affect NLR function. I wonder how the results would compare if smaller tags, like HA or FLAG, were used with both TaRx and CP.

Response: Thanks for the suggestion.

(1) We performed Y2H between CP and the mutated Ym1 alleles in the three mutants. The mutated Ym1 allele in YNXM-Mut2 encodes a truncated protein and did not interact with WYMV CP (Figure S9A). Co-infiltrating mutated Ym1 allele in YNXM-Mut2 and CP could not elicit cell death (Figure S9B). The mutated Ym1 alleles in YNXM-Mut1 and -Mut3 did interact with CP, however, neither of their interactions elicited cell death. Above results could explain why the three mutants are WYMV susceptible. We added this results in lines 308-314 in the revised manuscript.

(2) We performed cell death array by replacing the GFP tag with HA tag. We obtained similar results and revised Figure 4F in the revised manuscript.

Point 4: Figure 5E: The authors only used the CC domain to observe the impact of NLS and NES fusion on cell death, which may not fully reflect changes in TaRx activity. I suggest using full-length TaRx-NES and TaRx-NLS co-injected with CP to observe cell death. Additionally, as the authors have an infectious WYMV clone, I wonder if WYMV can infect *N. benthamiana*. If so, I also suggest testing the anti-WYMV activity of different TaRx derivatives, such as YNXM-Mut1-3 and TaRx-NES/NLS.

Response: Thanks for the suggestion.

(1) We performed cell death assay by co-infiltrating each of the full-length Ym1-NES and Ym1-NLS with CP into *N. benthamiana* leaves, and obtained similar results as before. Co-infiltration of Ym1-NES and CP elicit the cell death, while co-infiltration of Ym1-NLS and CP did not. We revised Figure 5 and described them accordingly in lines 325-331 in the revised manuscript.

(2) WYMV infectious clones can only cause systemic infection in wheat, but not in *N. benthamiana* (Zhang et al., 2021. <https://doi.org/10.1016/j.virol.2021.01.018>). Therefore, we directly tested the ability of cell death triggered by different Ym1 alleles with CP construct in *N. benthamiana* leaves. Our results demonstrated that only Ym1 can cause cell death, while its mutant alleles and derivative (e.g. TaRx-NLS) could not (Figure 5; Supplementary Figure 9).

Reviewer #3 (Remarks to the Author):

In this study, Chen et al. identified a wheat R gene named TaRx, which confers resistance to wheat yellow mosaic virus (WYMV). The authors describe TaRx as a typical CC-NLR protein that recognizes the coat protein of WYMV. Moreover, they suggest that the CC domain of TaRx functions as the cell death executor and that its cytoplasmic localization is involved in the cell death response. The discovery of the wheat R gene is important for global food security, and mechanistic insights into TaRx function are of broad interest within the plant biology research field. The quality of the data, data analyses, and figure descriptions—especially in the genetics section—are good and effectively support the authors' conclusions. I have a few comments on the functional analysis section. I have listed some comments that I hope will be useful for improving the manuscript:

Major comments:

Point 1: The name TaRx may be confusing, as researchers often refer to StRx from potato as Rx. Additionally, TaRx and StRx are not orthologous genes. I suggest the authors reconsider the gene name.

Response: Thanks for the suggestion. This is also suggested by Reviewer 2. We renamed TaRx as Ym1 (Wheat yellow mosaic virus resistance gene 1), and changed TaRx1 to Ym1 throughout the revised manuscript, Tables and Figures, supplementary materials and Responses to Editor and Reviewers.

Point 2: Line 285,286: The authors concluded that WYMV CP interacts weakly with full-length TaRx and strongly with the TaRx LRR domain. This difference may be due to varying protein expression levels in yeast. To support Figure 4B, the authors should perform western blot analyses to check the accumulation levels of full-length TaRx and each of its domains.

Response: Thanks for the comment. Yes, if we only tested them by Y2H in yeast system, these phenomena may be due to varying protein expression levels in yeast. In Figure 4C and 4D, the results from Co-IP and Luciferase assays can also support the point raised in the Figure 4B, that is WYMV CP interacts weakly with full-length Ym1 and strongly with the Ym1 LRR domain.

Point 3: Figure 5C-E: Testing the function of NES- and NLS-fused proteins is interesting; however, functional validation of NES- and NLS-fused full-length TaRx would be more informative than testing just the CC domain. I suggest that the authors assess the expression of TaRx-NES or TaRx-NLS with CP in both the cell death assay and cell biology experiment.

Response: Thanks for the suggestion. (1) We performed cell death assay by co-infiltrating each of the full-length Ym1-NES and Ym1-NLS and with CP into *N. benthamiana* leaves, and obtained similar results. Co-infiltration of Ym1-NES and CP elicit cell death while co-infiltration of Ym1-NLS and CP did not.

We revised Figure 5 and described them accordingly in lines 325-331 in the revised manuscript.

Minor comments:

1. Figure description of Figure 3D, E: I suggest clarifying which genes (TaRx or CP) are being detected in these figures.

Response: Thanks for the suggestion. We revised Figure 3D, E and added “Ym1 or CP probe”.

2. Line 288: I do not think that Co-IP and LCI assays detect the "physical interactions" between two proteins, as endogenous proteins in plants could mediate the association between them. The terms "interaction" or "association" would be more appropriate here.

Response: Thanks for the suggestion. We revised it accordingly in line 290 in the revised manuscript.

3. Figure 4F: It would be helpful to mention in the manuscript that TaRx/CP-induced cell death is weaker compared to CC and Bax, which is why trypan blue staining was conducted.

Response: Thanks for the suggestion. We revised it accordingly in line 297-299 in the revised manuscript.

4. Line 403-404: the authors should avoid mentioning the content of "data not shown."

Response: Thanks for the suggestion. We deleted them in the revised manuscript.

5. Figure legend of Figure 4 is not consistent with the current figure. Please revise this.

Response: Thanks for your carefulness and pointing it out. We revised the Figure legend of Figure 4 accordingly in the revised manuscript.